# Efficient Exploration and Value Function Generalization in Deterministic Systems

**Zheng Wen**
Stanford University
zhengwen@stanford.edu

**Benjamin Van Roy**
Stanford University
bvr@stanford.edu

## Abstract

We consider the problem of reinforcement learning over episodes of a finite-horizon deterministic system and as a solution propose *optimistic constraint propagation (OCP)*, an algorithm designed to synthesize efficient exploration and value function generalization. We establish that when the true value function $Q^*$ lies within the hypothesis class $\mathcal{Q}$, OCP selects optimal actions over all but at most $\dim_{\mathrm{E}}[\mathcal{Q}]$ episodes, where $\dim_{\mathrm{E}}$ denotes the *eluder dimension*. We establish further efficiency and asymptotic performance guarantees that apply even if $Q^*$ does not lie in $\mathcal{Q}$, for the special case where $\mathcal{Q}$ is the span of pre-specified indicator functions over disjoint sets.

## 1 Introduction

A growing body of work on efficient reinforcement learning provides algorithms with guarantees on sample and computational efficiency [13, 6, 2, 22, 4, 9]. This literature highlights the point that an effective exploration scheme is critical to the design of any efficient reinforcement learning algorithm. In particular, popular exploration schemes such as $\epsilon$-greedy, Boltzmann, and knowledge gradient can require learning times that grow exponentially in the number of states and/or the planning horizon.

The aforementioned literature focuses on *tabula rasa* learning; that is, algorithms aim to learn with little or no prior knowledge about transition probabilities and rewards. Such algorithms require learning times that grow at least linearly with the number of states. Despite the valuable insights that have been generated through their design and analysis, these algorithms are of limited practical import because state spaces in most contexts of practical interest are enormous. There is a need for algorithms that generalize from past experience in order to learn how to make effective decisions in reasonable time.

There has been much work on reinforcement learning algorithms that generalize (see, e.g., [5, 23, 24, 18] and references therein). Most of these algorithms do not come with statistical or computational efficiency guarantees, though there are a few noteworthy exceptions, which we now discuss. A number of results treat policy-based algorithms (see [10, 3] and references therein), in which the goal is to select high-performers among a pre-specified collection of policies as learning progresses. Though interesting results have been produced in this line of work, each entails quite restrictive assumptions or does not make strong guarantees. Another body of work focuses on model-based algorithms. An algorithm is proposed in [12] that fits a factored model to observed data and makes decisions based on the fitted model. The authors establish a sample complexity bound that is polynomial in the number of model parameters rather than the number of states, but the algorithm is computationally intractable because of the difficulty of solving factored MDPs. A recent paper [14] proposes a novel algorithm for the case where the true environment is known to belong to a finite or compact class of models, and shows that its sample complexity is polynomial in the cardinality of the model class if the model class is finite, or the $\epsilon$-covering-number if the

model class is compact. Though this result is theoretically interesting, for most model classes of interest, the $\epsilon$-covering-number is enormous since it typically grows exponentially in the number of free parameters. Another recent paper [17] establishes a regret bound for an algorithm that applies to problems with continuous state spaces and Hölder-continuous rewards and transition kernels. Though the results represent an interesting contribution to the literature, a couple features of the regret bound weaken its practical implications. First, regret grows linearly with the Hölder constant of the transition kernel, which for most contexts of practical relevance grows exponentially in the number of state variables. Second, the dependence on time becomes arbitrarily close to linear as the dimension of the state space grows. Reinforcement learning in linear systems with quadratic cost is treated in [1]. The method proposed is shown to realize regret that grows with the square root of time. The result is interesting and the property is desirable, but to the best of our knowledge, expressions derived for regret in the analysis exhibit an exponential dependence on the number of state variables, and further, we are not aware of a computationally efficient way of implementing the proposed method. This work was extended by [8] to address linear systems with sparse structure. Here, there are efficiency guarantees that scale gracefully with the number of state variables, but only under sparsity and other technical assumptions.

The most popular approach to generalization in the applied reinforcement learning literature involves fitting parameterized value functions. Such approaches relate closely to supervised learning in that they learn functions from state to value, though a difference is that value is influenced by action and observed only through delayed feedback. One advantage over model learning approaches is that, given a fitted value function, decisions can be made without solving a potentially intractable control problem. We see this as a promising direction, though there currently is a lack of theoretical results that provide attractive bounds on learning time with value function generalization. A relevant paper along this research line is [15], which studies the efficient reinforcement learning with value function generalization in the KWIK framework (see [16]), and reduces the efficient reinforcement learning problem to the efficient KWIK online regression problem. However, the authors do not show how to solve the general KWIK online regression problem efficiently, and it is not even clear whether this is possible. Thus, though the result of [15] is interesting, it does not provide a provably efficient algorithm.

An important challenge that remains is to couple exploration and value function generalization in a provably effective way, and in particular, to establish sample and computational efficiency guarantees that scale gracefully with the planning horizon and model complexity. In this paper, we aim to make progress in this direction. To start with a simple context, we restrict our attention to deterministic systems that evolve over finite time horizons, and we consider episodic learning, in which an agent repeatedly interacts with the same system. As a solution to the problem, we propose *optimistic constraint propagation (OCP)*, a computationally efficient reinforcement learning algorithm designed to synthesize efficient exploration and value function generalization. We establish that when the true value function $Q^*$ lies within the hypothesis class $\mathcal{Q}$, OCP selects optimal actions over all but at most $\dim_{\mathrm{E}}[\mathcal{Q}]$ episodes. Here, $\dim_{\mathrm{E}}$ denotes the *eluder dimension*, which quantifies complexity of the hypothesis class. A corollary of this result is that regret is bounded by a function that is constant over time and linear in the problem horizon and eluder dimension.

To put our aforementioned result in perspective, it is useful to relate it to other lines of work. Consider first the broad area of reinforcement learning algorithms that fit value functions, such as SARSA [19]. Even with the most commonly used sort of hypothesis class $\mathcal{Q}$, which is made up of linear combinations of fixed basis functions, and even when the hypothesis class contains the true value function $Q^*$, there are no guarantees that these algorithms will efficiently learn to make near-optimal decisions. On the other hand, our result implies that OCP attains near-optimal performance in time that scales linearly with the number of basis functions. Now consider the more specialized context of a deterministic linear system with quadratic cost and a finite time horizon. The analysis of [1] can be leveraged to produce regret bounds that scale exponentially in the number of state variables. On the other hand, using a hypothesis space $\mathcal{Q}$ consisting of quadratic functions of state-action pairs, the results of this paper show that OCP behaves near optimally within time that scales quadratically in the number of state and action variables.

We also establish efficiency and asymptotic performance guarantees that apply to agnostic reinforcement learning, where $Q^*$ does not necessarily lie in $\mathcal{Q}$. In particular, we consider the case where $\mathcal{Q}$ is the span of pre-specified indicator functions over disjoint sets. Our results here add to the literature on agnostic reinforcement learning with such a hypothesis class [21, 25, 7, 26]. Prior work in

this area has produced interesting algorithms and insights, as well as bounds on performance loss associated with potential limits of convergence, but no convergence or efficiency guarantees.

## 2 Reinforcement Learning in Deterministic Systems

In this paper, we consider an episodic reinforcement learning (RL) problem in which an agent repeatedly interacts with a discrete-time finite-horizon deterministic system, and refer to each interaction as an *episode*. The system is identified by a sextuple $\mathcal{M} = (\mathcal{S}, \mathcal{A}, H, F, R, S)$, where $\mathcal{S}$ is the state space, $\mathcal{A}$ is the action space, $H$ is the horizon, $F$ is a system function, $R$ is a reward function and $S$ is a sequence of states. If action $a \in \mathcal{A}$ is selected while the system is in state $x \in \mathcal{S}$ at period $t = 0, 1, \cdots, H - 1$, a reward of $R_t(x, a)$ is realized; furthermore, if $t < H - 1$, the state transitions to $F_t(x, a)$. Each episode terminates at period $H - 1$, and then a new episode begins. The initial state of episode $j$ is the $j$th element of $S$.

To represent the history of actions and observations over multiple episodes, we will often index variables by both episode and period. For example, $x_{j,t}$ and $a_{j,t}$ denote the state and action at period $t$ of episode $j$, where $j = 0, 1, \cdots$ and $t = 0, 1, \cdots, H - 1$. To count the total number of steps since the agent started learning, we say period $t$ in episode $j$ is time $jH + t$.

A (deterministic) policy $\mu = (\mu_0, \ldots, \mu_{H-1})$ is a sequence of functions, each mapping $\mathcal{S}$ to $\mathcal{A}$. For each policy $\mu$, define a value function $V_t^\mu(x) = \sum_{\tau=t}^{H-1} R_\tau(x_\tau, a_\tau)$, where $x_t = x$, $x_{\tau+1} = F_\tau(x_\tau, a_\tau)$, and $a_\tau = \mu_\tau(x_\tau)$. The optimal value function is defined by $V_t^*(x) = \sup_\mu V_t^\mu(x)$. A policy $\mu^*$ is said to be optimal if $V^{\mu^*} = V^*$. Throughout this paper, we will restrict attention to systems $\mathcal{M} = (\mathcal{S}, \mathcal{A}, H, F, R, S)$ that admit optimal policies. Note that this restriction incurs no loss of generality when the action space is finite.

It is also useful to define an action-contingent optimal value function: $Q_t^*(x, a) = R_t(x, a) + V_{t+1}^*(F_t(x, a))$ for $t < H - 1$, and $Q_{H-1}^*(x, a) = R_{H-1}(x, a)$. Then, a policy $\mu^*$ is optimal if $\mu_t^*(x) \in \arg\max_{a \in \mathcal{A}} Q_t^*(x, a)$ for all $(x, t)$.

A reinforcement learning algorithm generates each action $a_{j,t}$ based on observations made up to the $t$th period of the $j$th episode, including all states, actions, and rewards observed in previous episodes and earlier in the current episode, as well as the state space $\mathcal{S}$, action space $\mathcal{A}$, horizon $H$, and possible prior information. In each episode, the algorithm realizes reward $R^{(j)} = \sum_{t=0}^{H-1} R_t(x_{j,t}, a_{j,t})$. Note that $R^{(j)} \leq V_0^*(x_{j,0})$ for each $j$th episode. One way to quantify performance of a reinforcement learning algorithm is in terms of the number of episodes $J_L$ for which $R^{(j)} < V_0^*(x_{j,0}) - \epsilon$, where $\epsilon \geq 0$ is a pre-specified performance loss threshold. If the reward function $R$ is bounded, with $|R_t(x, a)| \leq \overline{R}$ for all $(x, a, t)$, then this also implies a bound on regret over episodes experienced prior to time $T$, defined by $\text{Regret}(T) = \sum_{j=0}^{\lfloor T/H \rfloor - 1} (V_0^*(x_{j,0}) - R^{(j)})$. In particular, $\text{Regret}(T) \leq 2\overline{R}HJ_L + \epsilon\lfloor T/H \rfloor$.

## 3 Optimistic Constraint Propagation

At a high level, our reinforcement learning algorithm – optimistic constraint propagation (OCP) – selects actions based on the *optimism in the face of uncertainty* principle and based on observed rewards and state transitions propagates constraints backwards through time. Specifically, it takes as input the state space $\mathcal{S}$, the action space $\mathcal{A}$, the horizon $H$, and a hypothesis class $\mathcal{Q}$ of candidates for $Q^*$. The algorithm maintains a sequence of subsets of $\mathcal{Q}$ and a sequence of scalar "upper bounds", which summarize constraints that past experience suggests for ruling out hypotheses. Each constraint in this sequence is specified by a state $x \in \mathcal{S}$, an action $a \in \mathcal{A}$, a period $t = 0, \ldots, H - 1$, and an interval $[L, U] \subseteq \Re$, and takes the form $\{Q \in \mathcal{Q} : L \leq Q_t(x, a) \leq U\}$. The upper bound of the constraint is $U$. Given a sequence $\mathcal{C} = (\mathcal{C}_1, \ldots, \mathcal{C}_{|\mathcal{C}|})$ of such constraints and upper bounds $\mathcal{U} = (\mathcal{U}_1, \ldots, \mathcal{U}_{|\mathcal{C}|})$, a set $\mathcal{Q}_{\mathcal{C}}$ is defined constructively by Algorithm 1. Note that if the constraints do not conflict then $\mathcal{Q}_{\mathcal{C}} = \mathcal{C}_1 \cap \cdots \cap \mathcal{C}_{|\mathcal{C}|}$. When constraints do conflict, priority is assigned first based on upper bound, with smaller upper bound preferred, and then, in the event of ties in upper bound, based on position in the sequence, with more recent experience preferred.

---
**Algorithm 1** Constraint Selection
---
**Require:** $\mathcal{Q}, \mathcal{C}$
  $\mathcal{Q}_{\mathcal{C}} \leftarrow \mathcal{Q}, u \leftarrow \min \mathcal{U}$
  **while** $u \leq \infty$ **do**
    **for** $\tau = |\mathcal{C}|$ to 1 **do**
      **if** $\mathcal{U}_{\tau} = u$ and $\mathcal{Q}_{\mathcal{C}} \cap \mathcal{C}_{\tau} \neq \varnothing$ **then**
        $\mathcal{Q}_{\mathcal{C}} \leftarrow \mathcal{Q}_{\mathcal{C}} \cap \mathcal{C}_{\tau}$
      **end if**
    **end for**
    **if** $\{u' \in \mathcal{U} : u' > u\} = \varnothing$ **then**
      **return** $\mathcal{Q}_{\mathcal{C}}$
    **end if**
    $u \leftarrow \min\{u' \in \mathcal{U} : u' > u\}$
  **end while**
---

OCP, presented below as Algorithm 2, at each time $t$ computes for the current state $x_{j,t}$ and each action $a$ the greatest state-action value $Q_t(x_{j,t}, a)$ among functions in $\mathcal{Q}_{\mathcal{C}}$ and selects an action that attains the maximum. In other words, an action is chosen based on the most optimistic feasible outcome subject to constraints. The subsequent reward and state transition give rise to a new constraint that is used to update $\mathcal{C}$. Note that the update of $\mathcal{C}$ is postponed until one episode is completed.

---
**Algorithm 2** Optimistic Constraint Propagation
---
**Require:** $\mathcal{S}$, $\mathcal{A}$, $H$, $\mathcal{Q}$
  Initialize $\mathcal{C} \leftarrow \varnothing$
  **for** episode $j = 0, 1, \cdots$ **do**
    Set $\mathcal{C}' \leftarrow \mathcal{C}$
    **for** period $t = 0, 1, \cdots, H-1$ **do**
      Apply $a_{j,t} \in \arg\max_{a \in \mathcal{A}} \sup_{Q \in \mathcal{Q}_{\mathcal{C}}} Q_t(x_{j,t}, a)$
      **if** $t < H-1$ **then**
        $U_{j,t} \leftarrow \sup_{Q \in \mathcal{Q}_{\mathcal{C}}} \left( R_t(x_{j,t}, a_{j,t}) + \sup_{a \in \mathcal{A}} Q_{t+1}(x_{j,t+1}, a) \right)$
        $L_{j,t} \leftarrow \inf_{Q \in \mathcal{Q}_{\mathcal{C}}} \left( R_t(x_{j,t}, a_{j,t}) + \sup_{a \in \mathcal{A}} Q_{t+1}(x_{j,t+1}, a) \right)$
      **else**
        $U_{j,t} \leftarrow R_t(x_{j,t}, a_{j,t})$
        $L_{j,t} \leftarrow R_t(x_{j,t}, a_{j,t})$
      **end if**
      $\mathcal{C}' \leftarrow \mathcal{C}' \frown \{Q \in \mathcal{Q} : L_{j,t} \leq Q_t(x_{j,t}, a_{j,t}) \leq U_{j,t}\}$
    **end for**
    Update $\mathcal{C} \leftarrow \mathcal{C}'$
  **end for**
---

Note that if $Q^* \in \mathcal{Q}$ then each constraint appended to $\mathcal{C}$ does not rule out $Q^*$, and therefore, the sequence of sets $\mathcal{Q}_{\mathcal{C}}$ generated as the algorithm progresses is decreasing and contains $Q^*$ in its intersection. In the agnostic case, where $Q^*$ may not lie in $\mathcal{Q}$, new constraints can be inconsistent with previous constraints, in which case selected previous constraints are relaxed as determined by Algorithm 1.

Let us briefly discuss several contexts of practical relevance and/or theoretical interest in which OCP can be applied.

- **Finite state/action tabula rasa case.** With finite state and action spaces, $Q^*$ can be represented as a vector, and without special prior knowledge, it is natural to let $\mathcal{Q} = \Re^{|\mathcal{S}| \cdot |\mathcal{A}| \cdot H}$.

- **Polytopic prior constraints.** Consider the aforementioned example, but suppose that we have prior knowledge that $Q^*$ lies in a particular polytope. Then we can let $\mathcal{Q}$ be that polytope and again apply OCP.

- **Linear systems with quadratic cost (LQ).** In this classical control model, if $\mathcal{S} = \Re^n$, $\mathcal{A} = \Re^m$, and $R$ is a positive semidefinite quadratic, then for each $t$, $Q_t^*$ is known to be a

positive semidefinite quadratic, and it is natural to let $\mathcal{Q} = \mathcal{Q}_0^H$ with $\mathcal{Q}_0$ denoting the set of positive semidefinite quadratics.

- **Finite hypothesis class.** Consider a context when we have prior knowledge that $Q^*$ can be well approximated by some element in a finite hypothesis class. Then we can let $\mathcal{Q}$ be that finite hypothesis class and apply OCP. This scenario is of particular interest from the perspective of learning theory. Note that this context entails agnostic learning, which is accommodated by OCP.

- **Linear combination of features.** It is often effective to hand-select a set of features $\phi_1, \ldots, \phi_K$, each mapping $\mathcal{S} \times \mathcal{A}$ to $\Re$, and, then for each $t$, aiming to compute weights $\theta^{(t)} \in \Re^K$ so that $\sum_k \theta_k^{(t)} \phi_k$ approximates $Q_t^*$ without knowing for sure that $Q_t^*$ lies in the span of the features. To apply OCP here, we would let $\mathcal{Q} = \mathcal{Q}_0^H$ with $\mathcal{Q}_0 = \mathrm{span}(\phi_1, \ldots, \phi_K)$. Note that this context also entails agnostic learning.

- **Sigmoid.** If it is known that rewards are only received upon transitioning to the terminal state and take values between $0$ and $1$, it might be appropriate to use a variation of the aforementioned feature based model that applies a sigmoidal function to the linear combination. In particular, we could have $\mathcal{Q} = \mathcal{Q}_0^H$ with $\mathcal{Q}_0 = \left\{ \psi\left(\sum_k \theta_k \phi_k(\cdot)\right) : \theta \in \Re^K \right\}$, where $\psi(z) = e^z/(1 + e^z)$.

It is worth mentioning that OCP, as we have defined it, assumes that an action $a$ maximizing $\sup_{Q \in \mathcal{Q}_c} Q_t(x_{j,t}, a)$ exists in each iteration. It is not difficult to modify the algorithm so that it addresses cases where this is not true. But we have not presented the more general form of OCP in order to avoid complicating this short paper.

## 4 Sample Efficiency of Optimistic Constraint Propagation

We now establish results concerning the sample efficiency of OCP. Our results bound the time it takes OCP to learn, and this must depend on the complexity of the hypothesis class. As such, we begin by defining the eluder dimension, as introduced in [20], which is the notion of complexity we will use.

### 4.1 Eluder Dimension

Let $\mathcal{Z} = \{(x, a, t) : x \in \mathcal{S}, a \in \mathcal{A}, t = 0, \ldots, H - 1\}$ be the set of all state-action-period triples, and let $\mathcal{Q}$ denote a nonempty set of functions mapping $\mathcal{Z}$ to $\Re$. For all $(x, a, t) \in \mathcal{Z}$ and $\tilde{\mathcal{Z}} \subseteq \mathcal{Z}$, $(x, a, t)$ is said to be *dependent* on $\tilde{\mathcal{Z}}$ with respect to $\mathcal{Q}$ if any pair of functions $Q, \tilde{Q} \in \mathcal{Q}$ that are equal on $\tilde{\mathcal{Z}}$ are equal at $(x, a, t)$. Further, $(x, a, t)$ is said to be *independent* of $\tilde{\mathcal{Z}}$ with respect to $\mathcal{Q}$ if $(x, a, t)$ is not dependent on $\tilde{\mathcal{Z}}$ with respect to $\mathcal{Q}$.

The *eluder dimension* $\dim_{\mathrm{E}}[\mathcal{Q}]$ of $\mathcal{Q}$ is the length of the longest sequence of elements in $\mathcal{Z}$ such that every element is independent of its predecessors. Note that $\dim_{\mathrm{E}}[\mathcal{Q}]$ can be zero or infinity, and it is straightforward to show that if $\mathcal{Q}_1 \subseteq \mathcal{Q}_2$ then $\dim_{\mathrm{E}}[\mathcal{Q}_1] \le \dim_{\mathrm{E}}[\mathcal{Q}_2]$. Based on results of [20], we can characterize the eluder dimensions of various hypothesis classes presented in the previous section.

- **Finite state/action tabula rasa case.** If $\mathcal{Q} = \Re^{|\mathcal{S}| \cdot |\mathcal{A}| \cdot H}$, then $\dim_{\mathrm{E}}[\mathcal{Q}] = |\mathcal{S}| \cdot |\mathcal{A}| \cdot H$.

- **Polytopic prior constraints.** If $\mathcal{Q}$ is a polytope of dimension $d$ in $\Re^{|\mathcal{S}| \cdot |\mathcal{A}| \cdot H}$, then $\dim_{\mathrm{E}}[\mathcal{Q}] = d$.

- **Linear systems with quadratic cost (LQ).** If $\mathcal{Q}_0$ is the set of positive semidefinite quadratics with domain $\Re^{m+n}$ and $\mathcal{Q} = \mathcal{Q}_0^H$, then $\dim_{\mathrm{E}}[\mathcal{Q}] = (m + n + 1)(m + n)H/2$.

- **Finite hypothesis space.** If $|\mathcal{Q}| < \infty$, then $\dim_{\mathrm{E}}[\mathcal{Q}] \le |\mathcal{Q}| - 1$.

- **Linear combination of features.** If $\mathcal{Q} = \mathcal{Q}_0^H$ with $\mathcal{Q}_0 = \mathrm{span}(\phi_1, \ldots, \phi_K)$, then $\dim_{\mathrm{E}}[\mathcal{Q}] \le KH$.

- **Sigmoid.** If $\mathcal{Q} = \mathcal{Q}_0^H$ with $\mathcal{Q}_0 = \left\{ \psi\left(\sum_k \theta_k \phi_k(\cdot)\right) : \theta \in \Re^K \right\}$, then $\dim_{\mathrm{E}}[\mathcal{Q}] \le KH$.

## 4.2 Learning with a Coherent Hypothesis Class

We now present results that apply when OCP is presented with a coherent hypothesis class; that is, where $Q^* \in \mathcal{Q}$. Our first result establishes that OCP can deliver less than optimal performance in no more than $\dim_E[\mathcal{Q}]$ episodes.

**Theorem 1** *For any system $\mathcal{M} = (\mathcal{S}, \mathcal{A}, H, F, R, S)$, if OCP is applied with $Q^* \in \mathcal{Q}$, then $|\{j : R^{(j)} < V_0^*(x_{j,0})\}| \leq \dim_E[\mathcal{Q}]$.*

This theorem follows from an "exploration-exploitation lemma", which asserts that in each episode, OCP either delivers optimal reward (exploitation) or introduces a constraint that reduces the eluder dimension of the hypothesis class by one (exploration). Consequently, OCP will experience sub-optimal performance in at most $\dim_E[\mathcal{Q}]$ episodes. A complete proof is provided in the appendix. An immediate corollary bounds regret.

**Corollary 1** *For any $\overline{R}$, any system $\mathcal{M} = (\mathcal{S}, \mathcal{A}, H, F, R, S)$ with $\sup_{(x,a,t)} |R_t(x,a)| \leq \overline{R}$, and any $T$, if OCP is applied with $Q^* \in \mathcal{Q}$, then $\mathrm{Regret}(T) \leq 2\overline{R}H\dim_E[\mathcal{Q}]$.*

Note the regret bound in Corollary 1 does not depend on time $T$, thus, it is an $O(1)$ bound. Furthermore, this regret bound is linear in $\overline{R}$, $H$ and $\dim_E[\mathcal{Q}]$, and does not directly depend on $|\mathcal{S}|$ or $|\mathcal{A}|$. The following results demonstrate that the bounds of the above theorem and corollary are sharp.

**Theorem 2** *For any reinforcement learning algorithm that takes as input a state space, an action space, a horizon, and a hypothesis class, there exists a system $\mathcal{M} = (\mathcal{S}, \mathcal{A}, H, F, R, S)$ and a hypothesis class $\mathcal{Q} \ni Q^*$ such that $|\{j : R^{(j)} < V_0^*(x_{j,0})\}| \geq \dim_E[\mathcal{Q}]$.*

**Theorem 3** *For any $\overline{R} \geq 0$ and any reinforcement learning algorithm that takes as input a state space, an action space, a horizon, and a hypothesis class, there exists a system $\mathcal{M} = (\mathcal{S}, \mathcal{A}, H, F, R, S)$ with $\sup_{(x,a,t)} |R_t(x,a)| \leq \overline{R}$ and a hypothesis class $\mathcal{Q} \ni Q^*$ such that $\sup_T \mathrm{Regret}(T) \geq 2\overline{R}H\dim_E[\mathcal{Q}]$.*

A constructive proof of these lower bounds is provided in the appendix. Following our discussion in previous sections, we discuss several interesting contexts in which the agent knows a coherent hypothesis class $\mathcal{Q}$ with finite eluder dimension.

- **Finite state/action tabula rasa case.** If we apply OCP in this case, then it will deliver sub-optimal performance in at most $|\mathcal{S}| \cdot |\mathcal{A}| \cdot H$ episodes. Furthermore, if $\sup_{(x,a,t)} |R_t(x,a)| \leq \overline{R}$, then for any $T$, $\mathrm{Regret}(T) \leq 2\overline{R}|\mathcal{S}||\mathcal{A}|H^2$.

- **Polytopic prior constraints.** If we apply OCP in this case, then it will deliver sub-optimal performance in at most $d$ episodes. Furthermore, if $\sup_{(x,a,t)} |R_t(x,a)| \leq \overline{R}$, then for any $T$, $\mathrm{Regret}(T) \leq 2\overline{R}Hd$.

- **Linear systems with quadratic cost (LQ).** If we apply OCP in this case, then it will deliver sub-optimal performance in at most $(m + n + 1)(m + n)H/2$ episodes.

- **Finite hypothesis class case.** Assume that the agent has prior knowledge that $Q^* \in \mathcal{Q}$, where $\mathcal{Q}$ is a finite hypothesis class. If we apply OCP in this case, then it will deliver sub-optimal performance in at most $|\mathcal{Q}|-1$ episodes. Furthermore, if $\sup_{(x,a,t)} |R_t(x,a)| \leq \overline{R}$, then for any $T$, $\mathrm{Regret}(T) \leq 2\overline{R}H[|\mathcal{Q}| - 1]$.

## 4.3 Agnostic Learning

As we have discussed in Section 3, OCP can also be applied in agnostic learning cases, where $Q^*$ may not lie in $\mathcal{Q}$. For such cases, the performance of OCP should depend on not only the complexity of $\mathcal{Q}$, but also the distance between $\mathcal{Q}$ and $Q^*$. We now present results when OCP is applied in a special agnostic learning case, where $\mathcal{Q}$ is the span of pre-specified indicator functions over disjoint subsets. We henceforth refer to this case as the state aggregation case.

Specifically, we assume that for any $t = 0, 1, \cdots, H - 1$, the state-action space at period $t$, $\mathcal{Z}_t = \{(x, a, t) : x \in \mathcal{S}, a \in \mathcal{A}\}$, can be partitioned into $K_t$ disjoint subsets $\mathcal{Z}_{t,1}, \mathcal{Z}_{t,2}, \cdots, \mathcal{Z}_{t,K_t}$, and use $\phi_{t,k}$ to denote the indicator function for partition $\mathcal{Z}_{t,k}$ (i.e. $\phi_{t,k}(x, a, t) = 1$ if $(x, a, t) \in \mathcal{Z}_{t,k}$, and $\phi_{t,k}(x, a, t) = 0$ otherwise). We define $K = \sum_{t=0}^{H-1} K_t$, and $\mathcal{Q}$ as

$$\mathcal{Q} = \mathrm{span}\left\{\phi_{0,1}, \phi_{0,2}, \cdots, \phi_{0,K_0}, \phi_{1,1}, \cdots, \phi_{H-1,K_{H-1}}\right\}. \tag{4.1}$$

Note that $\dim_{\mathrm{E}}[\mathcal{Q}] = K$. We define the distance between $Q^*$ and the hypothesis class $\mathcal{Q}$ as

$$\rho = \min_{Q \in \mathcal{Q}} \|Q - Q^*\|_\infty = \min_{Q \in \mathcal{Q}} \sup_{(x,a,t)} |Q_t(x, a) - Q_t^*(x, a)|. \tag{4.2}$$

The following result establishes that with $\mathcal{Q}$ and $\rho$ defined above, the performance loss of OCP is larger than $2\rho H(H + 1)$ in at most $K$ episodes.

**Theorem 4** *For any system $\mathcal{M} = (\mathcal{S}, \mathcal{A}, H, F, R, S)$, if OCP is applied with $\mathcal{Q}$ defined in Eqn(4.1), then*

$$|\{j : R^{(j)} < V_0^*(x_{j,0}) - 2\rho H(H + 1)\}| \leq K,$$

*where $K$ is the number of partitions and $\rho$ is defined in Eqn(4.2).*

Similar to Theorem 1, this theorem also follows from an "exploration-exploitation lemma", which asserts that in each episode, OCP either delivers near-optimal reward (exploitation), or approximately determines $Q_t^*(x, a)$'s for all the $(x, a, t)$'s in a disjoint subset (exploration). A complete proof for Theorem 4 is provided in the appendix. An immediate corollary bounds regret.

**Corollary 2** *For any $\overline{R} \geq 0$, any system $\mathcal{M} = (\mathcal{S}, \mathcal{A}, H, F, R, S)$ with $\sup_{(x,a,t)} |R_t(x, a)| \leq \overline{R}$, and any time $T$, if OCP is applied with $\mathcal{Q}$ defined in Eqn(4.1), then $\mathrm{Regret}(T) \leq 2\overline{R}KH + 2\rho(H + 1)T$, where $K$ is the number of partitions and $\rho$ is defined in Eqn(4.2).*

Note that the regret bound in Corollary 2 is $O(T)$, and the coefficient of the linear term is $2\rho(H+1)$. Consequently, if $Q^*$ is close to $\mathcal{Q}$, then the regret will increase slowly with $T$. Furthermore, the regret bound in Corollary 2 does not directly depend on $|\mathcal{S}|$ or $|\mathcal{A}|$.

We further notice that the threshold performance loss in Theorem 4 is $O(\rho H^2)$. The following proposition provides a condition under which the performance loss in one episode is $O(\rho H)$.

**Proposition 1** *For any episode $j$, if $\forall t = 0, 1, \cdots, H - 1$,*

$$\mathcal{Q}_\mathcal{C} \subseteq \{Q \in \mathcal{Q} : L_{j,t} \leq Q_t(x_{j,t}, a_{j,t}) \leq U_{j,t}\},$$

*then we have $V_0^*(x_{j,0}) - R^{(j)} \leq 6\rho H = O(\rho H)$.*

That is, if all the new constraints in an episode are redundant, then the performance loss in that episode is $O(\rho H)$. Note that if the condition for Proposition 1 holds in an episode, then $\mathcal{Q}_\mathcal{C}$ will not be modified at the end of that episode. Furthermore, if the system has a fixed initial state and the condition for Proposition 1 holds in one episode, then it will hold in all the subsequent episodes, and consequently, the performance losses in all the subsequent episodes are $O(\rho H)$.

## 5 Computational Efficiency of Optimistic Constraint Propagation

We now briefly discuss the computational complexity of OCP. As typical in the complexity analysis of optimization algorithms, we assume that basic operations include the arithmetic operations, comparisons, and assignment, and measure computational complexity in terms of the number of basic operations (henceforth referred to as operations) per period.

First, it is worth pointing out that for a general hypothesis class $\mathcal{Q}$ and general action space $\mathcal{A}$, the per period computations of OCP can intractable. This is because:

- Computing $\sup_{Q \in \mathcal{Q}_\mathcal{C}} Q_t(x_{j,t}, a)$, $U_{j,t}$ and $L_{j,t}$ requires solving a possibly intractable optimization problems.

- Selecting an action that maximizes $\sup_{Q \in \mathcal{Q}_\mathcal{C}} Q_t(x_{j,t}, a)$ can be intractable.

Further, the number of constraints in $\mathcal{C}$, and with it the number of operations per period, can grow over time.

However, if $|\mathcal{A}|$ is tractably small and $\mathcal{Q}$ has some special structures (e.g. $\mathcal{Q}$ is a finite set or a linear subspace or, more generally a polytope), then by discarding some "redundant" constraints in $\mathcal{C}$, OCP with a variant of Algorithm 1 will be computationally efficient, and the sample efficiency results developed in Section 4 will still hold. Due to space limitations, we only discuss the scenario where $\mathcal{Q}$ is a polytope of dimension $d$. Note that the finite state/action tabula rasa case, the linear-quadratic case, and the case with linear combinations of disjoint indicator functions are all special cases of this scenario.

Specifically, if $\mathcal{Q}$ is a polytope of dimension $d$ (i.e., within a $d$-dimensional subspace), then any $Q \in \mathcal{Q}$ can be represented by a weight vector $\theta \in \Re^d$, and $\mathcal{Q}$ can be characterized by a set of linear inequalities of $\theta$. Furthermore, the new constraints of the form $L_{j,t} \leq Q_t(x_{j,t}, a_{j,t}) \leq U_{j,t}$ are also linear inequalities of $\theta$. Hence, in each episode, $\mathcal{Q}_\mathcal{C}$ is characterized by a polyhedron in $\Re^d$, and $\sup_{Q \in \mathcal{Q}_\mathcal{C}} Q_t(x_{j,t}, a)$, $U_{j,t}$ and $L_{j,t}$ can be computed by solving linear programming (LP) problems. If we assume that all the encountered numerical values can be represented with $B$ bits, and LPs are solved by Karmarkar's algorithm [11], then the following proposition bounds the computational complexity.

**Proposition 2** *If $\mathcal{Q}$ is a polytope of dimension $d$, each numerical value in the problem data or observed in the course of learning can be represented with $B$ bits, and OCP uses Karmarkar's algorithm to solve linear programs, then the computational complexity of OCP is $O\left([|\mathcal{A}| + |\mathcal{C}|]\, |\mathcal{C}| d^{4.5} B\right)$ operations per period.*

The proof of Proposition 2 is provided in the appendix. Notice that the computational complexity is polynomial in $d$, $B$, $|\mathcal{C}|$ and $|\mathcal{A}|$, and thus, OCP will be computationally efficient if all these parameters are tractably small. Note that the bound in Proposition 2 is a worst-case bound, and the $O(d^{4.5})$ term is incurred by the need to solve LPs. For some special cases, the computational complexity is much less. For instance, in the state aggregation case, the computational complexity is $O(|\mathcal{C}| + |\mathcal{A}| + d)$ operations per period.

As we have discussed above, one can ensure that $|\mathcal{C}|$ remains bounded by using variants of Algorithm 1 that discard the redundant constraints and/or update $\mathcal{Q}_\mathcal{C}$ more efficiently. Specifically, it is straightforward to design such constraint selection algorithms if $\mathcal{Q}$ is a coherent hypothesis class, or if $\mathcal{Q}$ is the span of pre-specified indicator functions over disjoint sets. Furthermore, if the notion of redundant constraints is properly defined, the sample efficiency results derived in Section 4 will still hold.

# 6 Conclusion

We have proposed a novel reinforcement learning algorithm, called optimistic constraint propagation (OCP), that synthesizes efficient exploration and value function generalization for reinforcement learning in deterministic systems. We have shown that OCP is sample efficient if $Q^*$ lies in the given hypothesis class, or if the given hypothesis class is the span of pre-specified indicator functions over disjoint sets.

It is worth pointing out that for more general reinforcement learning problems, how to design provably sample efficient algorithms with value function generalization is currently still open. For instance, it is not clear how to establish such algorithms for the general agnostic learning case discussed in this paper, as well as for reinforcement learning in MDPs. One interesting direction for future research is to extend OCP, or a variant of it, to these two problems.

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
