[Supplementary Material]

# Efficient Exploration and Value Function Generalization in Deterministic Systems: Appendices

Zheng Wen　　　　　Benjamin Van Roy

Before proceeding, we first define some useful terminologies and notations. Specifically, to distinguish $\mathcal{C}$ in different episodes, we use $\mathcal{C}_j$ to denote the $\mathcal{C}$ in episode $j$. For simplicity of notations, we use $z$ as a shorthand notation for a state-action-time triple $(x, a, t)$, and use $\mathcal{Z}$ to denote the set of all state-action-time triples, that is

$$\mathcal{Z} = \{(x, a, t) : x \in \mathcal{S},\ a \in \mathcal{A},\ \text{and}\ t = 0, 1, \cdots, H - 1\}.$$

## Appendix A    Proofs for Theorem 1 and Corollary 1

We prove Theorem 1 and Corollary 1 in this section. First of all, the following lemma states that $Q^* \in \mathcal{Q}_{\mathcal{C}_j}$, $\forall j = 0, 1, \cdots$.

**Lemma 1** *For any $j = 0, 1, \cdots$, we have (1) $Q^* \in \mathcal{Q}_{\mathcal{C}_j}$ and (2) $L_{j,t} \leq Q_t^*(x_{j,t}, a_{j,t}) \leq U_{j,t}$, $\forall t = 0, \cdots, H-1$.*

**Proof:**
First, we prove that $\forall j$, if $Q^* \in \mathcal{Q}_{\mathcal{C}_j}$, then $L_{j,t} \leq Q_t^*(x_{j,t}, a_{j,t}) \leq U_{j,t}$, $\forall t = 0, 1, \cdots, H - 1$. To see it, notice that for $t = H - 1$, by definition, we have $Q_t^*(x_{j,t}, a_{j,t}) = R_t(x_{j,t}, a_{j,t})$. Furthermore, from the OCP algorithm, we also have $L_{j,t} = U_{j,t} = R_t(x_{j,t}, a_{j,t})$. Thus $L_{j,t} \leq Q_t^*(x_{j,t}, a_{j,t}) \leq U_{j,t}$ trivially holds. On the other hand, for any $t < H - 1$, from Bellman equation, we have

$$Q_t^*(x_{j,t}, a_{j,t}) = R_t(x_{j,t}, a_{j,t}) + V_{t+1}^*(x_{j,t+1}),$$

thus, from the definitions of $L_{j,t}$ and $U_{j,t}$, $L_{j,t} \leq Q_t^*(x_{j,t}, a_{j,t}) \leq U_{j,t}$ if and only if

$$\inf_{Q \in \mathcal{Q}_{\mathcal{C}_j}} \sup_{a \in \mathcal{A}} Q_{t+1}(x_{j,t+1}, a) \leq V_{t+1}^*(x_{j,t+1}) \leq \sup_{Q \in \mathcal{Q}_{\mathcal{C}_j}} \sup_{a \in \mathcal{A}} Q_{t+1}(x_{j,t+1}, a).$$

Note that $V_{t+1}^*(x_{j,t+1}) = \sup_{a \in \mathcal{A}} Q_{t+1}^*(x_{j,t+1}, a)$, thus, the above inequality trivially holds since $Q^* \in \mathcal{Q}_{\mathcal{C}_j}$.

We now prove $Q^* \in \mathcal{Q}_{\mathcal{C}_j}$, $\forall j$ by induction on $j$. First, notice that if $j = 0$, by definition, $\mathcal{Q}_{\mathcal{C}_0} = \mathcal{Q}$. From the definition of the coherent hypothesis class, we have $Q^* \in \mathcal{Q} = \mathcal{Q}_{\mathcal{C}_0}$. Now assume for any $j' < j$, we have $Q^* \in \mathcal{Q}_{\mathcal{C}_{j'}}$. Thus, from our analysis above, we have

$$L_{j',t'} \leq Q_{t'}^*(x_{j',t'}, a_{j',t'}) \leq U_{j',t'}, \quad \forall j' < j, \forall t = 0, 1, \cdots, H - 1.$$

Since we also have $Q^* \in \mathcal{Q}$, thus, there is no inconsistent constraint before episode $j$. Thus, from the constraint selection algorithm, we have

$$\mathcal{Q}_{\mathcal{C}_j} = \mathcal{Q} \cap \left[ \cap_{j' < j} \cap_{t=0,1,\cdots,H-1} \{Q : L_{j',t'} \leq Q_{t'}(x_{j',t'}, a_{j',t'}) \leq U_{j',t'}\} \right],$$

and hence $Q^* \in \mathcal{Q}_{\mathcal{C}_j}$. By mathematical induction, we have $Q^* \in \mathcal{Q}_{\mathcal{C}_j}$, $\forall j$.

---
**Algorithm 3** Definition of $\mathcal{Z}_j$ and $t_j^*$
---
Initialize $\mathcal{Z}_0 \leftarrow \emptyset$
**for** $j = 0, 1, \cdots$ **do**
   Set $t_j^* \leftarrow$ NULL
   **if** $\exists t = 0, 1, \cdots, H-1$ s.t. $(x_{j,t}, a_{j,t}, t)$ is marginally independent of $\mathcal{Z}_j$ with respect to $\mathcal{Q}$ **then**
     Set

$$t_j^* \quad \leftarrow \quad \text{last period } t \text{ in episode } j \text{ s.t. } (x_{j,t}, a_{j,t}, t) \text{ is marginally}$$
$$\text{independent of } \mathcal{Z}_j \text{ with respect to } \mathcal{Q}$$

    and

$$\mathcal{Z}_{j+1} \leftarrow \left[ \mathcal{Z}_j, (x_{j,t_j^*}, a_{j,t_j^*}, t_j^*) \right]$$

   **else**
     Set $\mathcal{Z}_{j+1} \leftarrow \mathcal{Z}_j$
   **end if**
**end for**
---

Combining the above two results, we also have $L_{j,t} \leq Q_t^*(x_{j,t}, a_{j,t}) \leq U_{j,t}$, $\forall (j,t)$. **q.e.d.**

Before proceeding, we define two more useful notations. Specifically, for any episode $j = 0, 1, \cdots$, we define $\mathcal{Z}_j$ and $t_j^*$ by Algorithm 3.

Note that based on the definition, $\forall j = 0, 1, \cdots$,

- $\mathcal{Z}_j$ is a sequence (ordered set) of elements in $\mathcal{Z}$. Furthermore, each element in $\mathcal{Z}_j$ is marginally independent of its predecessors.

- If $t_j^* \neq$ NULL, then it is the last period in episode $j$ s.t. $(x_{j,t}, a_{j,t}, t)$ is marginally independent of $\mathcal{Z}_j$ with respect to $\mathcal{Q}$.

Based on the notions of $\mathcal{Z}_j$ and $t_j^*$, we can prove the following technical lemma:

**Lemma 2** $\forall j = 0, 1, \cdots$ and $\forall t = 0, 1, \cdots, T-1$, we have

(a) $\forall z \in \mathcal{Z}_j$ and $\forall Q \in \mathcal{Q}_{\mathcal{C}_j}$, we have $Q(z) = Q^*(z)$.

(b) If $(x_{j,t}, a_{j,t}, t)$ is marginally dependent on $\mathcal{Z}_j$ with respect to $\mathcal{Q}$, then (1) $a_{j,t}$ is optimal and (2)

$$Q_t(x_{j,t}, a_{j,t}) = Q_t^*(x_{j,t}, a_{j,t}) = V_t^*(x_{j,t}), \quad \forall Q \in \mathcal{Q}_{\mathcal{C}_j}.$$

**Proof:**
We prove this lemma by induction on $j$. First, notice that if $j = 0$, then from Algorithm 3, we have $\mathcal{Z}_0 = \emptyset$. Thus, Lemma 2(a) holds for $j = 0$.

Second, we prove that if Lemma 2(a) holds for episode $j$, then Lemma 2(b) holds for episode $j$ and Lemma 2(a) holds for episode $j + 1$. To see why Lemma 2(b) holds for episode $j$, notice that from Lemma 1, we have $Q^* \in \mathcal{Q}_{\mathcal{C}_j} \subseteq \mathcal{Q}$. Furthermore, from the induction hypothesis, we have

$$Q(z) = Q^*(z), \quad \forall z \in \mathcal{Z}_j \text{ and } \forall Q \in \mathcal{Q}_{\mathcal{C}_j} \subseteq \mathcal{Q}.$$

Since $(x_{j,t}, a_{j,t}, t)$ is marginally dependent on $\mathcal{Z}_j$ with respect to $\mathcal{Q}$, then we have that

$$Q_t(x_{j,t}, a_{j,t}) = Q_t^*(x_{j,t}, a_{j,t}), \quad \forall Q \in \mathcal{Q}_{\mathcal{C}_j} \subseteq \mathcal{Q}.$$

Hence we have $\sup_{Q \in \mathcal{Q}_{\mathcal{C}_j}} Q_t(x_{j,t}, a_{j,t}) = Q_t^*(x_{j,t}, a_{j,t})$, furthermore, from the OCP algorithm, we have $\sup_{Q \in \mathcal{Q}_{\mathcal{C}_j}} Q_t(x_{j,t}, a_{j,t}) \geq \sup_{Q \in \mathcal{Q}_{\mathcal{C}_j}} Q_t(x_{j,t}, a)$, $\forall a \in \mathcal{A}$, thus we have

$$Q_t^*(x_{j,t}, a_{j,t}) = \sup_{Q \in \mathcal{Q}_{\mathcal{C}_j}} Q_t(x_{j,t}, a_{j,t}) \geq \sup_{Q \in \mathcal{Q}_{\mathcal{C}_j}} Q_t(x_{j,t}, a) \geq Q_t^*(x_j, a), \quad \forall a \in \mathcal{A},$$

where the last inequality follows from the fact that $Q^* \in \mathcal{Q}_{\mathcal{C}_j}$. Thus, $a_{j,t}$ is optimal and $Q_t^*(x_{j,t}, a_{j,t}) = V_t^*(x_{j,t})$. Thus, Lemma 2(b) holds for episode $j$.

We now prove Lemma 2(a) holds for episode $j+1$. We prove the conclusion by considering two different scenarios. If $t_j^* = \text{NULL}$, then $\mathcal{Z}_{j+1} = \mathcal{Z}_j$ and $\mathcal{Q}_{\mathcal{C}_{j+1}} \subseteq \mathcal{Q}_{\mathcal{C}_j}$. Thus, obviously, Lemma 2(a) holds for episode $j+1$. On the other hand, if $t_j^* \neq \text{NULL}$, we have $\mathcal{Q}_{\mathcal{C}_{j+1}} \subseteq \mathcal{Q}_{\mathcal{C}_j}$ and

$$\mathcal{Z}_{j+1} = \left[ \mathcal{Z}_j, (x_{j,t_j^*}, a_{j,t_j^*}, t_j^*) \right].$$

Based on the induction hypothesis, $\forall z \in \mathcal{Z}_j$ and $\forall Q \in \mathcal{Q}_{\mathcal{C}_{j+1}} \subseteq \mathcal{Q}_{\mathcal{C}_j}$, we have $Q(z) = Q^*(z)$. Thus, it is sufficient to prove that

$$Q_{t_j^*}(x_{j,t_j^*}, a_{j,t_j^*}) = Q_{t_j^*}^*(x_{j,t_j^*}, a_{j,t_j^*}), \quad \forall Q \in \mathcal{Q}_{\mathcal{C}_{j+1}}. \tag{1}$$

We prove Eqn(1) by considering two different cases. First, if $t_j^* = H - 1$, it is sufficient to prove that

$$Q_{H-1}(x_{j,H-1}, a_{j,H-1}) = R_{H-1}(x_{j,H-1}, a_{j,H-1}), \quad \forall Q \in \mathcal{Q}_{\mathcal{C}_{j+1}},$$

which holds by definition of $\mathcal{Q}_{\mathcal{C}_{j+1}}$ (see OCP algorithm, and recall from Lemma 1 that no constraints are conflicting). On the other hand, if $t_j^* < H - 1$, it is sufficient to prove that

$$Q_{t_j^*}(x_{j,t_j^*}, a_{j,t_j^*}) = R_{t_j^*}(x_{j,t_j^*}, a_{j,t_j^*}) + V_{t_j^*+1}^*(x_{j,t_j^*+1}), \quad \forall Q \in \mathcal{Q}_{\mathcal{C}_{j+1}}.$$

Recall that OCP algorithm add a constraint $L_{j,t_j^*} \leq Q_{t_j^*}(x_{j,t_j^*}, a_{j,t_j^*}) \leq U_{j,t_j^*}$ to $\mathcal{Q}_{\mathcal{C}_{j+1}}$ (and again, from Lemma 1, no constraints are conflicting). Based on the definitions of $L_{j,t_j^*}$ and $U_{j,t_j^*}$, it is sufficient to prove that

$$V_{t_j^*+1}^*(x_{j,t_j^*+1}) = \sup_{Q \in \mathcal{Q}_{\mathcal{C}_j}} \sup_{a \in \mathcal{A}} Q_{t_j^*+1}(x_{j,t_j^*+1}, a) = \inf_{Q \in \mathcal{Q}_{\mathcal{C}_j}} \sup_{a \in \mathcal{A}} Q_{t_j^*+1}(x_{j,t_j^*+1}, a). \tag{2}$$

We first prove that

$$V_{t_j^*+1}^*(x_{j,t_j^*+1}) = \sup_{Q \in \mathcal{Q}_{\mathcal{C}_j}} \sup_{a \in \mathcal{A}} Q_{t_j^*+1}(x_{j,t_j^*+1}, a). \tag{3}$$

Specifically, we have that

$$\sup_{Q \in \mathcal{Q}_{\mathcal{C}_j}} \sup_{a \in \mathcal{A}} Q_{t_j^*+1}(x_{j,t_j^*+1}, a) = \sup_{a \in \mathcal{A}} \sup_{Q \in \mathcal{Q}_{\mathcal{C}_j}} Q_{t_j^*+1}(x_{j,t_j^*+1}, a) = \sup_{Q \in \mathcal{Q}_{\mathcal{C}_j}} Q_{t_j^*+1}(x_{j,t_j^*+1}, a_{j,t_j^*+1}) = V_{t_j^*+1}^*(x_{j,t_j^*+1}),$$

where

- The second equality follows from the fact that $a_{j,t_j^*+1} \in \arg\max_{a \in \mathcal{A}} \sup_{Q \in \mathcal{Q}_{\mathcal{C}_j}} Q_{t_j^*+1}(x_{j,t_j^*+1}, a)$.

- The last equality follows from the definition of $t_j^*$ and Part (b) of the lemma for episode $j$ (which we have just proved above, and holds by the induction hypothesis). Specifically, since $t_j^*$ is the last period in episode $j$ s.t. $(x_{j,t}, a_{j,t}, t)$ is marginally independent of $\mathcal{Z}_j$ with respect to $\mathcal{Q}$. Thus, $(x_{j,t_j^*+1}, a_{j,t_j^*+1}, t_j^*+1)$ is marginally dependent on $\mathcal{Z}_j$ with respect to $\mathcal{Q}$. From Lemma 2(b) for episode $j$, we have $V_{t_j^*+1}^*(x_{j,t_j^*+1}) = Q_{t_j^*+1}(x_{j,t_j^*+1}, a_{j,t_j^*+1})$ for any $Q \in \mathcal{Q}_{\mathcal{C}_j}$. Thus, $\sup_{Q \in \mathcal{Q}_{\mathcal{C}_j}} Q_{t_j^*+1}(x_{j,t_j^*+1}, a_{j,t_j^*+1}) = V_{t_j^*+1}^*(x_{j,t_j^*+1})$.

On the other hand, we have that

$$\inf_{Q \in \mathcal{Q}_{\mathcal{C}_j}} \sup_{a \in \mathcal{A}} Q_{t_j^*+1}(x_{j,t_j^*+1}, a) \geq \sup_{a \in \mathcal{A}} \inf_{Q \in \mathcal{Q}_{\mathcal{C}_j}} Q_{t_j^*+1}(x_{j,t_j^*+1}, a) \geq \inf_{Q \in \mathcal{Q}_{\mathcal{C}_j}} Q_{t_j^*+1}(x_{j,t_j^*+1}, a_{j,t_j^*+1}) = V_{t_j^*+1}^*(x_{j,t_j^*+1}),$$

where the first inequality follows from the max-min inequality, and the second inequality follows from the fact that $a_{j,t_j^*+1} \in \mathcal{A}$. Recall that we have $V_{t_j^*+1}^*(x_{j,t_j^*+1}) = Q_{t_j^*+1}(x_{j,t_j^*+1}, a_{j,t_j^*+1})$ for any $Q \in \mathcal{Q}_{\mathcal{C}_j}$. Thus, $\inf_{Q \in \mathcal{Q}_{\mathcal{C}_j}} Q_{t_j^*+1}(x_{j,t_j^*+1}, a_{j,t_j^*+1}) = V_{t_j^*+1}^*(x_{j,t_j^*+1})$ and the last equality holds. Hence we have

$$V_{t_j^*+1}^*(x_{j,t_j^*+1}) = \sup_{Q \in \mathcal{Q}_{\mathcal{C}_j}} \sup_{a \in \mathcal{A}} Q_{t_j^*+1}(x_{j,t_j^*+1}, a) \geq \inf_{Q \in \mathcal{Q}_{\mathcal{C}_j}} \sup_{a \in \mathcal{A}} Q_{t_j^*+1}(x_{j,t_j^*+1}, a) \geq V_{t_j^*+1}^*(x_{j,t_j^*+1}).$$

Thus, Eqn(2) holds. Hence, Lemma 2(a) holds for episode $j + 1$.

Thus, by induction, we have proved Lemma 2. **q.e.d.**

Based on Lemma 2, we prove the following exploration/exploitation lemma, which states that in each episode $j$, OCP algorithm will either achieve the optimal reward (exploitation), or update $\mathcal{Q}_{\mathcal{C}_{j+1}}$ based on the Q-value at a marginally independent state-action-time triple (exploration).

**Lemma 3** *For any $j = 0, 1, \cdots$, we have*

- **Exploration:** If $t_j^* \neq$ NULL, then $(x_{j,t_j^*}, a_{j,t_j^*}, t_j^*)$ is marginally independent of $\mathcal{Z}_j$ and $|\mathcal{Z}_{j+1}| = |\mathcal{Z}_j| + 1$. Furthermore, $\forall Q \in \mathcal{Q}_{\mathcal{C}_{j+1}}$, we have $Q_{t_j^*}(x_{j,t_j^*}, a_{j,t_j^*}) = Q_{t_j^*}^*(x_{j,t_j^*}, a_{j,t_j^*})$.

- **Exploitation:** If $t_j^* =$ NULL, then $R^{(j)} = V_0^*(x_{j,0})$.

**Proof:**
Note that from Algorithm 3, if $t_j^* =$ NULL, then $\forall t = 0, 1, \cdots, T - 1$, $(x_{j,t}, a_{j,t}, t)$ is marginally dependent on $\mathcal{Z}_j$ with respect to $\mathcal{Q}$. Thus, from Lemma 2(b), $a_{j,t}$ is optimal $\forall t = 0, 1, \cdots, H - 1$. Hence we have

$$R^{(j)} = \sum_{t=0}^{H-1} R_t(x_{j,t}, a_{j,t}) = V_0^*(x_{j,0}).$$

On the other hand, $t_j^* \neq$ NULL, then from Algorithm 3, $(x_{j,t_j^*}, a_{j,t_j^*}, t_j^*)$ is marginally independent of $\mathcal{Z}_j$ and $|\mathcal{Z}_{j+1}| = |\mathcal{Z}_j| + 1$. Note $(x_{j,t_j^*}, a_{j,t_j^*}, t_j^*) \in \mathcal{Z}_{j+1}$, hence from Lemma 2(a), $\forall Q \in \mathcal{Q}_{\mathcal{C}_{j+1}}$, we have $Q_{t_j^*}(x_{j,t_j^*}, a_{j,t_j^*}) = Q_{t_j^*}^*(x_{j,t_j^*}, a_{j,t_j^*})$. **q.e.d.**

We now prove Theorem 1 based on Lemma 3.

**Proof for Theorem 1:**
Notice that $\forall j = 0, 1, \cdots$, $R^{(j)} \leq V_0^*(x_{j,0})$ by definition. Thus, from Lemma 3, $R^{(j)} < V_0^*(x_{j,0})$ implies that $t_j^* \neq$ NULL. Hence we have

$$\mathbf{1}\left[R^{(j)} < V_0^*(x_{j,0})\right] \leq \mathbf{1}\left[t_j^* \neq \text{NULL}\right], \quad \forall j = 0, 1, \cdots.$$

Furthermore, notice that from the definition of $\mathcal{Z}_j$, we have

$$\mathbf{1}\left[t_j^* \neq \text{NULL}\right] = |\mathcal{Z}_{j+1}| - |\mathcal{Z}_j|,$$

where $|\cdot|$ denotes the length of the given sequence. Notice that both $|\mathcal{Z}_{j+1}|$ and $|\mathcal{Z}_j|$ are finite, since by definition, we have $|\mathcal{Z}_j| \leq j$ for any $j = 0, 1, \cdots$. Thus for any $J = 0, 1, \cdots$, we have

$$\sum_{j=0}^{J} \mathbf{1}\left[R^{(j)} < V_0^*(x_{j,0})\right] \leq \sum_{j=0}^{J} \mathbf{1}\left[t_j^* \neq \text{NULL}\right] = \sum_{j=0}^{J}[|\mathcal{Z}_{j+1}| - |\mathcal{Z}_j|] = |\mathcal{Z}_{J+1}| - |\mathcal{Z}_0| = |\mathcal{Z}_{J+1}|, \qquad (4)$$

where the last equality follows from the fact that $|\mathcal{Z}_0| = |\emptyset| = 0$.

Notice that by definition (see Algorithm 3), $\forall j = 0, 1, \cdots$, $\mathcal{Z}_j$ is a sequence of elements in $\mathcal{Z}$ such that every element is marginally independent of its predecessors with respect to $\mathcal{Q}$. Hence, from the definition of margin dimension, we have $|\mathcal{Z}_j| \leq \dim_{\text{M}}[\mathcal{Q}]$, $\forall j = 0, 1, \cdots$. Combining this result with Eqn(4), we have

$$\sum_{j=0}^{J} \mathbf{1}\left[R^{(j)} < V_0^*(x_{j,0})\right] \leq |\mathcal{Z}_{J+1}| \leq \dim_{\text{M}}[\mathcal{Q}], \quad \forall J = 0, 1, \cdots. \qquad (5)$$

Finally, notice that $\sum_{j=0}^{J} \mathbf{1}\left[V_j < V_0^*(x_{j,0})\right]$ is a non-decreasing function of $J$, and is bounded above by $\dim_{\text{M}}[\mathcal{Q}]$. Thus,

$$\lim_{J \to \infty} \sum_{j=0}^{J} \mathbf{1}\left[R^{(j)} < V_0^*(x_{j,0})\right] = \sum_{j=0}^{\infty} \mathbf{1}\left[R^{(j)} < V_0^*(x_{j,0})\right]$$

exists, and satisfies

$$\sum_{j=0}^{\infty} \mathbf{1}\left[R^{(j)} < V_0^*(x_{j,0})\right] \leq \dim_{\text{M}}[\mathcal{Q}].$$

Hence we have $\left|\left\{j : R^{(j)} < V_0^*(x_{j,0})\right\}\right| \leq \dim_{\text{M}}[\mathcal{Q}]$. **q.e.d.**

We now prove Corollary 1 based on Theorem 1:

**Proof for Corollary 1:**
Notice that if $R^{(j)} = V_0^*(x_{j,0})$, then $V_0^*(x_{j,0}) - R^{(j)} = 0$; on the other hand, if $R^{(j)} < V_0^*(x_{j,0})$, then we have

$$V_0^*(x_{j,0}) - R^{(j)} \leq 2\overline{R}H,$$

since the maximum achievable total reward is $\overline{R}H$, while the minimum achievable total reward is $-\overline{R}H$. Hence we have the following key inequality:

$$V_0^*(x_{j,0}) - V_j \leq 2\overline{R}H\mathbf{1}\left[R^{(j)} < V_0^*(x_{j,0})\right], \quad \forall j = 0, 1, \cdots.$$

Thus, $\forall J \geq 0$, we have

$$\sum_{j=0}^{J}\left[V_0^*(x_{j,0}) - R^{(j)}\right] \leq 2\overline{R}H \sum_{j=0}^{J} \mathbf{1}\left[R^{(j)} < V_0^*(x_{j,0})\right] \leq 2\overline{R}H\dim_{\text{M}}[\mathcal{Q}],$$

where the second inequality follows from Eqn(5) in the proof for Theorem 1.

Recall that by definition, for any $T$, we have

$$\text{Regret}(T) = \sum_{j=0}^{\lfloor T/H \rfloor - 1} (V_0^*(x_{j,0}) - R^{(j)}) \leq 2\overline{R}H\dim_{\text{M}}[\mathcal{Q}].$$

**q.e.d.**

# Appendix B  Proofs for Theorem 2 and 3

We provide a constructive proof for Theorem 2 and 3. Before proceeding, we first define some useful terminologies and notations. First, for any state space $\mathcal{S}$, any time horizon $H = 1, 2, \cdots$, any action space $\mathcal{A}$, and any function class $\mathcal{Q}$, we use $\mathbb{M}(\mathcal{S}, \mathcal{A}, H, \mathcal{Q})$ to denote the set of all finite-horizon MDP $\mathcal{M}$'s satisfying the following conditions:

1. The state space of $\mathcal{M}$ is $\mathcal{S}$.

2. The time horizon of $\mathcal{M}$ is $H$.

3. The single action space of $\mathcal{M}$ is $\mathcal{A}$.

4. $\mathcal{M}$ admits an optimal policy $\mu^*$.

5. $Q^*$, the optimal Q-function of $\mathcal{M}$, belongs to the function class $\mathcal{Q}$.

Notice that for any reinforcement learning algorithm that takes $\mathcal{S}$, $\mathcal{A}$, $H$, $\mathcal{Q}$ as input, and knows that $\mathcal{Q}$ is a coherent hypothesis class, $\mathbb{M}(\mathcal{S}, \mathcal{A}, H, \mathcal{Q})$ is the set of all finite-horizon MDPs that are consistent with the algorithm's prior information.

We prove a result that is stronger than Theorem 2 and 3 by considering a scenario in which an adversary adaptively chooses an MDP model $\mathcal{M} \in \mathbb{M}(\mathcal{S}, \mathcal{A}, H, \mathcal{Q})$. Specifically, we assume that

- At the beginning of each episode $j$, the adversary adaptively chooses the initial state $x_{j,0}$ for that episode.

- At period $t$ in episode $j$, the agent first chooses an action $a_{j,t} \in \mathcal{A}$ based on some RL algorithm[1], and then the adversary adaptively chooses a set of state-action-time triples $\mathcal{Z}_{j,t} \subseteq \mathcal{Z}$ and specifies the instantaneous rewards and state transitions on $\mathcal{Z}_{j,t}$, subject to the constraints that (1) $(x_{j,t}, a_{j,t}, t) \in \mathcal{Z}_{j,t}$ and (2) these adaptively specified instantaneous rewards and state transitions must be consistent with the agent's prior knowledge and past observations.

We assume that the adversary's objective is to maximize the number of episodes in which the agent achieves sub-optimal rewards. Then we have the following lemma:

**Lemma 4** $\forall H = 1, 2, \cdots, \forall K = 1, 2, \cdots,$ and $\forall \overline{R} \geq 0$, there exist a state space $\mathcal{S}$, an action space $\mathcal{A}$ and a function class $\mathcal{Q}$ with $\dim_{\mathrm{M}}[\mathcal{Q}] = K$ such that no matter how the agent adaptively chooses actions, the adversary can adaptively choose a finite-horizon MDP $\mathcal{M} \in \mathbb{M}(\mathcal{S}, \mathcal{A}, H, \mathcal{Q})$ satisfying the following conditions:

- $\sup_{(x,a)} |R_t(x,a)| \leq \overline{R}$.

- The agent will achieve sub-optimal rewards in at least $K$ episodes, and $\sup_T \mathrm{Regret}(T) \geq 2\overline{R}HK$.

**Proof:**
We provide a constructive proof for Lemma 4. Specifically, $\forall H = 1, 2, \cdots, \forall K = 1, 2, \cdots,$ and $\forall \overline{R} \geq 0$, we construct the state space as $\mathcal{S} = \{1, 2, \cdots, 2K\}$, and the action space as $\mathcal{A} = \{1, 2\}$. Recall that $\mathcal{Z} = \{(x, a, t) : x \in \mathcal{S}, t = 0, 1, \cdots, H-1, \text{ and } a \in \mathcal{A}\}$, thus, for $\mathcal{S}$ and $\mathcal{A}$ constructed above, we have $|\mathcal{Z}| = 4KH$. Hence, $Q^*$, the optimal Q-function, can be represented as a vector in $\Re^{4KH}$.

Figure 1: Illustration of the state transition

Before constructing the function class $\mathcal{Q}$, we first define a matrix $\Phi \in \Re^{4KH \times K}$ as follows. $\forall (x, a, t) \in \mathcal{Z}$, let $\Phi(x, a, t) \in \Re^K$ denote the row of $\Phi$ corresponding to the state-action-time triple $(x, a, t)$, we construct $\Phi(x, a, t)$ as:

$$\Phi(x, a, t) = \begin{cases} (T - t)\mathbf{e}_k & \text{if } x = 2k - 1 \text{ for some } k = 1, \cdots, K, \ a = 1, 2 \text{ and } t = 1, \cdots, H - 1 \\ -(T - t)\mathbf{e}_k & \text{if } x = 2k \text{ for some } k = 1, \cdots, K, \ a = 1, 2 \text{ and } t = 1, \cdots, H - 1 \\ T\mathbf{e}_k & \text{if } x = 2k - 1 \text{ or } 2k \text{ for some } k = 1, \cdots, K, \ a = 1 \text{ and } t = 0 \\ -T\mathbf{e}_k & \text{if } x = 2k - 1 \text{ or } 2k \text{ for some } k = 1, \cdots, K, \ a = 2 \text{ and } t = 0 \end{cases} \quad (6)$$

Notice that $\text{rank}(\Phi) = K$. We construct $\mathcal{Q} = \text{span}\,[\Phi]$, thus we have

$$\dim_\text{M}[\mathcal{Q}] = \dim_\text{M}[\text{span}\,[\Phi]] = \dim\,(\text{span}\,[\Phi]) = \text{rank}(\Phi) = K.$$

Now we describe how the adversary adaptively chooses a finite-horizon MDP $\mathcal{M} \in \mathbb{M}\,(\mathcal{S}, \mathcal{A}, T, \mathcal{Q})$:

- For any $j = 0, 1, \cdots$, at the beginning of episode $j$, the adversary chooses the initial state in that episode as

$$x_{j,0} = (j \bmod K) \times 2 + 1.$$

That is, $x_{0,0} = x_{K,0} = x_{2K,0} = \cdots = 1$, $x_{1,0} = x_{K+1,0} = x_{2K+1,0} = \cdots = 3$ ...

- Before interacting with the agent, the adversary chooses the following system function $F$[2]:

$$F_t(x, a) = \begin{cases} 2k - 1 & \text{if } t = 0, \ x = 2k - 1 \text{ or } 2k \text{ for some } k = 1, \cdots, K, \text{ and } a = 1 \\ 2k & \text{if } t = 0, \ x = 2k - 1 \text{ or } 2k \text{ for some } k = 1, \cdots, K, \text{ and } a = 2 \\ x & \text{if } t = 1, \cdots, H - 2 \text{ and } a = 1, 2 \end{cases} \quad .$$

The state transition is illustrated in Figure 1.

- In episode $j = 0, 1, \cdots, K - 1$, the adversary adaptively chooses the reward function $R$ as follows. If the agent takes action 1 in period 0 in episode $j$ at initial state $x_{j,0} = 2j + 1$, then the adversary set

$$R_0(2j+1,1) = R_0(2j+2,1) = R_t(2j+1,1) = R_t(2j+1,2) = -\overline{R}$$
$$R_0(2j+1,2) = R_0(2j+2,2) = R_t(2j+2,1) = R_t(2j+2,2) = \overline{R}$$

$\forall t = 1, 2, \cdots, H - 1$. Otherwise (i.e. if the agent takes action 2 in period 0 in episode $j$), then the adversary set

$$R_0(2j+1,1) = R_0(2j+2,1) = R_t(2j+1,1) = R_t(2j+1,2) = \overline{R}$$
$$R_0(2j+1,2) = R_0(2j+2,2) = R_t(2j+2,1) = R_t(2j+2,2) = -\overline{R}$$

Notice that the adversary completes the construction of the MDP model $\mathcal{M}$ at the end of episode $K - 1$.

We now prove that the constructed MDP model $\mathcal{M} \in \mathbb{M}(\mathcal{S}, \mathcal{A}, T, \mathcal{Q})$. This is sufficient to prove $Q^*$, the optimal Q-function of $\mathcal{M}$, lies in the function class $\mathcal{Q}$. To see it, notice that $\forall j = 0, 1, \cdots, K - 1$, if the agent takes action $a_{j,0} = 1$ in period 0 in episode $j$, then from the constructed MDP model $\mathcal{M}$, we have

$$Q_0^*(2j+1,1) = Q_0^*(2j+2,1) = -\overline{R}H$$
$$Q_0^*(2j+1,2) = Q_0^*(2j+2,2) = \overline{R}H$$
$$Q_t^*(2j+1,a) = -\overline{R}(H-t) \quad \forall t = 1, \cdots, H-1, \forall a = 1, 2$$
$$Q_t^*(2j+2,a) = \overline{R}(H-t) \quad \forall t = 1, \cdots, H-1, \forall a = 1, 2$$

On the other hand, if the agent takes action $a_{j,0} = 2$ in period 0 in episode $j$, then we have

$$Q_0^*(2j+1,1) = Q_0^*(2j+2,1) = \overline{R}H$$
$$Q_0^*(2j+1,2) = Q_0^*(2j+2,2) = -\overline{R}H$$
$$Q_t^*(2j+1,a) = \overline{R}(H-t) \quad \forall t = 1, \cdots, H-1, \forall a = 1, 2$$
$$Q_t^*(2j+2,a) = -\overline{R}(H-t) \quad \forall t = 1, \cdots, H-1, \forall a = 1, 2$$

Note that $(a_{0,0}, a_{1,0}, \cdots, a_{K-1,0})$ completely determines the constructed MDP model $\mathcal{M}$. For the convenience of exposition, we use $\mathcal{M}(a_{0,0}, a_{1,0}, \cdots, a_{K-1,0})$ to denote this particular MDP model.

Recall that $\mathcal{Q} = \text{span}[\Phi]$, where $\Phi$ is defined in Eqn(6). Note that based on the definition of $\Phi$, for any combination of

$$(a_{0,0}, a_{1,0}, \cdots, a_{K-1,0}) \in \{1, 2\}^K,$$

the optimal Q-function of MDP $\mathcal{M}(a_{0,0}, a_{1,0}, \cdots, a_{K-1,0})$ lies in $\mathcal{Q}$. Specifically, the optimal Q-function of MDP $\mathcal{M}(a_{0,0}, a_{1,0}, \cdots, a_{K-1,0})$ is $\Phi\theta$, where $\theta \in \Re^K$, and $\theta_k$, the $k$th element of $\theta$, is defined as

$$\theta_k = \begin{cases} -\overline{R} & \text{if } a_{k-1} = 1 \\ \overline{R} & \text{if } a_{k-1} = 2 \end{cases},$$

for any $k = 1, 2, \cdots, K$.

Finally, we show that the constructed MDP model $\mathcal{M}$ satisfies Lemma 4. First, notice that obviously, we have $|R_t(x,a)| \leq \overline{R}$, $\forall (x, a, t) \in \mathcal{Z}$. Second, we note that the agent achieves sub-optimal rewards in the first $K$ episodes, thus, he will achieve sub-optimal rewards in at least $K$ episodes. The cumulative regret in the first $K$ episodes is $2KH\overline{R}$, thus, $\sup_T \text{Regret}(T) \geq 2KH\overline{R}$.

**q.e.d.**

Since the fact that an adversary can adaptively choose a "bad" MDP model simply implies that such MDP model exists, thus, Theorem 2 and 3 follow from Lemma 4.

# Appendix C  Proofs for Theorem 4, Corollary 2 and Proposition 1

Before proceeding, we first briefly how constraint selection algorithm updates $\mathcal{Q}_{\mathcal{C}}$'s for the function class $\mathcal{Q}$ specified in Eqn(4.1). Specifically, let $\theta_{t,k}$ denote the coefficient of the indicator function $\phi_{t,k}$, $\forall(t,k)$. Assume that $(x, a, t)$ belongs to partition $\mathcal{Z}_{t,k}$, then, with $\mathcal{Q}$ specified in Eqn(4.1), $L \leq Q_t(x, a) \leq U$ is a constraint on and only on $\theta_{t,k}$, and is equivalent to $L \leq \theta_{t,k} \leq U$. By induction, it is straightforward to see in episode $j$, $\mathcal{Q}_{\mathcal{C}_j}$ can be represented as

$$\left\{ \theta \in \Re^K : \underline{\theta}_{t,k}^{(j)} \leq \theta_{t,k} \leq \overline{\theta}_{t,k}^{(j)}, \; \forall(t,k) \right\},$$

for some $\underline{\theta}_{t,k}^{(j)}$'s and $\overline{\theta}_{t,k}^{(j)}$'s. Note that $\underline{\theta}_{t,k}^{(j)}$ can be $-\infty$ and $\overline{\theta}_{t,k}^{(j)}$ can be $\infty$, and, when $j = 0$, $\overline{\theta}_{t,k}^{(0)} = \infty$ and $\underline{\theta}_{t,k}^{(0)} = -\infty$. Furthermore, from the constraint selection algorithm, $\overline{\theta}_{t,k}^{(j)}$ is monotonically non-increasing in $j$, for any $(t, k)$ (since when ranking the constraints, constraints with smaller upper bound are preferred). Specifically, if OCP adds a new constraint $L \leq \theta_{t,k} \leq U$ on $\theta_{t,k}$ in episode $j$, we have $\overline{\theta}_{t,k}^{(j+1)} = \min\{\overline{\theta}_{t,k}^{(j)}, U\}$; otherwise, $\overline{\theta}_{t,k}^{(j+1)} = \overline{\theta}_{t,k}^{(j)}$. Thus, if $\overline{\theta}_{t,k}^{(j)} < \infty$, then $\forall j' \geq j$, we have $\overline{\theta}_{t,k}^{(j')} < \infty$.

For any $(x, a, t) \in \mathcal{Z}$, and any $j$, we define the optimistic Q-function in episode $j$, $Q_{j,t}^{\copyright}(x, a)$ as

$$Q_{j,t}^{\copyright}(x, a) = \sup_{Q \in \mathcal{Q}_{\mathcal{C}_j}} Q_t(x, a),$$

and the pessimistic Q-function in episode $j$, $Q_{j,t}^{\copyright}(x, a)$ as

$$Q_{j,t}^{\copyright}(x, a) = \inf_{Q \in \mathcal{Q}_{\mathcal{C}_j}} Q_t(x, a).$$

Clearly, if $(x, a, t) \in \mathcal{Z}_{t,k}$, then we have $Q_{j,t}^{\copyright}(x, a) = \overline{\theta}_{t,k}^{(j)}$, and $Q_{j,t}^{\copyright}(x, a) = \underline{\theta}_{t,k}^{(j)}$. Moreover, $(x, a, t)$'s in the same partition have the same optimistic and pessimistic Q-values.

It is also worth pointing out that by definition of $\rho$, if $(x, a, t)$ and $(x', a', t)$ are in the same partition, then we have $|Q_t^*(x, a) - Q_t^*(x', a')| \leq 2\rho$. To see it, let $\tilde{Q} \in \arg\min_{Q \in \mathcal{Q}} \|Q - Q^*\|_\infty$, then we have $|\tilde{Q}_t(x, a) - Q_t^*(x, a)| \leq \rho$ and $|\tilde{Q}_t(x', a') - Q_t^*(x', a')| \leq \rho$. Since $\tilde{Q} \in \mathcal{Q}$ and $(x, a, t)$ and $(x', a', t)$ are in the same partition, we have $\tilde{Q}_t(x, a) = \tilde{Q}_t(x', a')$. Then from triangular inequality, we have $|Q_t^*(x, a) - Q_t^*(x', a')| \leq 2\rho$.

We first prove the following lemma:

**Lemma 5** $\forall(x, a, t)$ and $\forall j = 0, 1, \cdots$, if $Q_{j,t}^{\copyright}(x, a) < \infty$, then $|Q_{j,t}^{\copyright}(x, a) - Q_t^*(x, a)| \leq 2\rho(H - t)$.

**Proof:**
We prove Lemma 5 by induction on $j$. Note that when $j = 0$, $\forall(x, a, t)$, $Q_{j,t}^{\copyright}(x, a) = \infty$. Thus, Lemma 5 trivially holds for $j = 0$.

Second, we prove that if Lemma 5 holds for episode $j$, then it also holds for episode $j + 1$, for any $j = 0, 1, \cdots$. To prove this result, it is sufficient to show that for any $(x, a, t)$ whose associated optimistic Q-value has been updated in episode $j$ (i.e. $Q_{j,t}^{\copyright}(x, a) \neq Q_{j+1,t}^{\copyright}(x, a)$), if the new optimistic Q-value $Q_{j+1,t}^{\copyright}(x, a)$ is still finite, then we have

$$|Q_{j+1,t}^{\copyright}(x, a) - Q_t^*(x, a)| \leq 2\rho(H - t).$$

This is because for any $(x, a, t)$ with $Q_{j+1,t}^{\copyright}(x, a) = Q_{j,t}^{\copyright}(x, a)$, Lemma 5 holds for episode $j + 1$ by induction hypothesis.

Note that if $Q^{\odot}_{j,t}(x,a) \neq Q^{\odot}_{j+1,t}(x,a)$, then $(x,a,t)$ must be in the same partition $\mathcal{Z}_{t,k}$ as $(x_{j,t}, a_{j,t}, t)$. Furthermore, from the discussion above, and noting that $\sup_{Q \in \mathcal{Q}_{\mathcal{C}_j}} \sup_{a \in \mathcal{A}} Q_{t+1}(x_{j,t+1}, a) = \sup_{a \in \mathcal{A}} Q^{\odot}_{j,t+1}(x_{j,t+1}, a)$, we have

$$Q^{\odot}_{j+1,t}(x,a) = \overline{\theta}^{(j+1)}_{t,k} = \begin{cases} R_{H-1}(x_{j,H-1}, a_{j,H-1}) & \text{if } t = H-1 \\ R_t(x_{j,t}, a_{j,t}) + \sup_{a \in \mathcal{A}} Q^{\odot}_{j,t+1}(x_{j,t+1}, a) & \text{if } t < H-1 \end{cases}$$

We now prove $|Q^{\odot}_{j+1,t}(x,a) - Q^*_t(x,a)| \leq 2\rho(H-t)$ by considering two different scenarios:

- If $t = H-1$, note that $Q^{\odot}_{j+1,t}(x,a) = R_{H-1}(x_{j,H-1}, a_{j,H-1}) = Q^*_{H-1}(x_{j,H-1}, a_{j,H-1})$, since $(x,a,t)$ and $(x_{j,H-1}, a_{j,H-1}, H-1)$ are in the same partition, from our discussion above, we have $|Q^*_t(x,a) - Q^*_{H-1}(x_{j,H-1}, a_{j,H-1})| \leq 2\rho$. Hence we have that

$$|Q^*_t(x,a) - Q^{\odot}_{j+1,t}(x,a)| \leq 2\rho = 2\rho(H-t).$$

- If $t < H-1$, note that

$$Q^{\odot}_{j+1,t}(x,a) = R_t(x_{j,t}, a_{j,t}) + \sup_{a \in \mathcal{A}} Q^{\odot}_{j,t+1}(x_{j,t+1}, a).$$

If $Q^{\odot}_{j+1,t}(x,a) < \infty$, then $\sup_{a \in \mathcal{A}} Q^{\odot}_{j,t+1}(x_{j,t+1}, a) < \infty$, and hence $Q^{\odot}_{j,t+1}(x_{j,t+1}, a) < \infty$, $\forall a \in \mathcal{A}$. Furthermore, from the induction hypothesis, $Q^{\odot}_{j,t+1}(x_{j,t+1}, a) < \infty$, $\forall a \in \mathcal{A}$, implies that

$$\left| Q^{\odot}_{j,t+1}(x_{j,t+1}, a) - Q^*_{t+1}(x_{j,t+1}, a) \right| \leq 2\rho(H-t-1), \quad \forall a \in \mathcal{A}. \tag{7}$$

On the other hand, from the Bellman equation at $(x_{j,t}, a_{j,t}, t)$, we have that

$$Q^*_t(x_{j,t}, a_{j,t}) = R_t(x_{j,t}, a_{j,t}) + V^*_{t+1}(x_{j,t+1}) = R_t(x_{j,t}, a_{j,t}) + \sup_{a \in \mathcal{A}} Q^*_{t+1}(x_{j,t+1}, a).$$

Consequently, we have that

$$\begin{aligned} \left| Q^{\odot}_{j+1,t}(x,a) - Q^*_t(x_{j,t}, a_{j,t}) \right| &= \left| \sup_{a \in \mathcal{A}} Q^{\odot}_{j,t+1}(x_{j,t+1}, a) - \sup_{a \in \mathcal{A}} Q^*_{t+1}(x_{j,t+1}, a) \right| \\ &\leq \sup_{a \in \mathcal{A}} \left| Q^{\odot}_{j,t+1}(x_{j,t+1}, a) - Q^*_{t+1}(x_{j,t+1}, a) \right| \\ &\leq 2\rho(H-t-1). \end{aligned} \tag{8}$$

On the other hand, since $(x,a,t)$ and $(x_{j,t}, a_{j,t}, t)$ are in the same partition, we have

$$|Q^*_t(x,a) - Q^*_t(x_{j,t}, a_{j,t})| \leq 2\rho,$$

consequently, we have

$$\left| Q^{\odot}_{j+1,t}(x,a) - Q^*_t(x,a) \right| \leq 2\rho(H-t).$$

Thus, Lemma 5 holds for episode $j+1$. By induction, we have proved Lemma 5. **q.e.d.**

Based on Lemma 5, we have the following result:

**Lemma 6** $\forall j = 0, 1, \cdots$, if $Q_{j,t}^{\odot}(x_{j,t}, a_{j,t}) < \infty$ for any $t = 0, 1, \cdots, H - 1$, then we have

$$V_0^*(x_{j,0}) - R^{(j)} \leq 2\rho H(H+1) = O\left(\rho H^2\right). \tag{9}$$

Furthermore, if the conditions of Proposition 1 hold, then we have

$$V_0^*(x_{j,0}) - R^{(j)} \leq 6\rho H = O(\rho H). \tag{10}$$

**Proof:**

Notice that from OCP algoriothm, $\forall t = 0, 1, \cdots, H - 1$, we have

$$Q_{j,t}^{\odot}(x_{j,t}, a_{j,t}) \geq Q_{j,t}^{\odot}(x_{j,t}, a), \quad \forall a \in \mathcal{A}.$$

Thus, if $Q_{j,t}^{\odot}(x_{j,t}, a_{j,t}) < \infty$ for any $t = 0, 1, \cdots, H - 1$, we then have

$$Q_{j,t}^{\odot}(x_{j,t}, a) < \infty, \quad \forall t = 0, 1, \cdots, H - 1 \text{ and } \forall a \in \mathcal{A}.$$

Consequently, from Lemma 5, we have that

$$\left| Q_t^*(x_{j,t}, a) - Q_{j,t}^{\odot}(x_{j,t}, a) \right| \leq 2\rho(H - t), \quad \forall t = 0, 1, \cdots, H - 1 \text{ and } \forall a \in \mathcal{A}.$$

Thus, for any $t = 0, 1, \cdots, H - 1$, we have

$$Q_t^*(x_{j,t}, a_{j,t}) + 2\rho(H - t) \geq Q_{j,t}^{\odot}(x_{j,t}, a_{j,t}) \geq Q_{j,t}^{\odot}(x_{j,t}, a) \geq Q_t^*(x_{j,t}, a) - 2\rho(H - t), \quad \forall a \in \mathcal{A}.$$

Hence we have $Q_t^*(x_{j,t}, a_{j,t}) \geq Q_t^*(x_{j,t}, a) - 4\rho(H - t)$, $\forall a \in \mathcal{A}$. Thus we have

$$Q_t^*(x_{j,t}, a_{j,t}) \geq \sup_{a \in \mathcal{A}} Q_t^*(x_{j,t}, a) - 4\rho(H - t) = V_t^*(x_{j,t}) - 4\rho(H - t). \tag{11}$$

Notice that the above inequality holds for any $t = 0, 1, \cdots, H - 1$.

We first prove Eqn(9). Note that from Bellman equation, we have

$$Q_t^*(x_{j,t}, a_{j,t}) = \begin{cases} R_t(x_{j,t}, a_{j,t}) + V_{t+1}^*(x_{j,t+1}) & \text{if } t < H - 1 \\ R_{H-1}(x_{j,H-1}, a_{j,H-1}) & \text{if } t = H - 1 \end{cases}$$

Thus, for any $t < H - 1$, we have

$$R_t(x_{j,t}, a_{j,t}) \geq V_t^*(x_{j,t}) - V_{t+1}^*(x_{j,t+1}) - 4\rho(H - t),$$

and

$$R_{H-1}(x_{j,H-1}, a_{j,H-1}) \geq V_{H-1}^*(x_{j,H-1}) - 4\rho.$$

Summing up the above inequalities, we have

$$\sum_{t=0}^{H-1} R_t(x_{j,t}, a_{j,t}) \geq V_0^*(x_{j,0}) - \sum_{t=0}^{H-1} [4\rho(H - t)] = V_0^*(x_{j,0}) - 2\rho H(H+1).$$

That is $V_0^*(x_{j,0}) - \sum_{t=0}^{H-1} R_t(x_{j,t}, a_{j,t}) = V_0^*(x_{j,0}) - R^{(j)} \leq 2\rho H(H+1)$.

We now prove Eqn(10). Specifically, for any $t = 0, 1, \cdots, H-1$, if

$$\mathcal{Q}_{\mathcal{C}_j} \subseteq \{Q \in \mathcal{Q} : L_{j,t} \leq Q_t(x_{j,t}, a_{j,t}) \leq U_{j,t}\},$$

then we will have

$$U_{j,t} \geq Q_{j,t}^{\smiley}(x_{j,t}, a_{j,t}) \geq Q_{j,t}^{\frownie}(x_{j,t}, a_{j,t}) \geq L_{j,t}.$$

Note that by definition, $U_{j,H-1} = L_{j,H-1} = R_{H-1}(x_{j,H-1}, a_{j,H-1})$, and for $t < H-1$, we have

$$U_{j,t} = R_t(x_{j,t}, a_{j,t}) + \sup_{a \in \mathcal{A}} Q_{j,t+1}^{\smiley}(x_{j,t+1}, a) = R_t(x_{j,t}, a_{j,t}) + Q_{j,t+1}^{\smiley}(x_{j,t+1}, a_{j,t+1}),$$

and

$$
\begin{aligned}
L_{j,t} &= R_t(x_{j,t}, a_{j,t}) + \inf_{Q \in \mathcal{Q}_{\mathcal{C}_j}} \sup_{a \in \mathcal{A}} Q_{t+1}(x_{j,t+1}, a) \\
&\geq R_t(x_{j,t}, a_{j,t}) + \sup_{a \in \mathcal{A}} Q_{j,t+1}^{\frownie}(x_{j,t+1}, a) \geq R_t(x_{j,t}, a_{j,t}) + Q_{j,t+1}^{\frownie}(x_{j,t+1}, a_{j,t+1}),
\end{aligned}
$$

where the first inequality follows from the max-min inequality, and the second inequality follows from the fact that $a_{j,t+1} \in \mathcal{A}$. Thus we have

$$
\begin{aligned}
Q_{j,t}^{\frownie}(x_{j,t}, a_{j,t}) &\geq R_t(x_{j,t}, a_{j,t}) + Q_{j,t+1}^{\frownie}(x_{j,t+1}, a_{j,t+1}) \quad \forall t < H-1 \\
Q_{j,H-1}^{\frownie}(x_{j,H-1}, a_{j,H-1}) &\geq R_{H-1}(x_{j,H-1}, a_{j,H-1}).
\end{aligned}
$$

Thus, we have $Q_{j,0}^{\frownie}(x_{j,0}, a_{j,0}) \geq \sum_{t=0}^{H-1} R_t(x_{j,t}, a_{j,t}) = R^{(j)}$. Similarly, we have that

$$
\begin{aligned}
Q_{j,t}^{\smiley}(x_{j,t}, a_{j,t}) &\leq R_t(x_{j,t}, a_{j,t}) + Q_{j,t+1}^{\smiley}(x_{j,t+1}, a_{j,t+1}) \quad \forall t < H-1 \\
Q_{j,H-1}^{\smiley}(x_{j,H-1}, a_{j,H-1}) &\leq R_{H-1}(x_{j,H-1}, a_{j,H-1}),
\end{aligned}
$$

and hence $Q_{j,0}^{\smiley}(x_{j,0}, a_{j,0}) \leq \sum_{t=0}^{H-1} R_t(x_{j,t}, a_{j,t}) = R^{(j)}$. So we have

$$Q_{j,t}^{\smiley}(x_{j,t}, a_{j,t}) \geq Q_{j,t}^{\frownie}(x_{j,t}, a_{j,t}) \geq R^{(j)} \geq Q_{j,0}^{\smiley}(x_{j,0}, a_{j,0}),$$

and hence

$$Q_{j,0}^{\smiley}(x_{j,0}, a_{j,0}) = Q_{j,0}^{\frownie}(x_{j,0}, a_{j,0}) = R^{(j)}.$$

Since $Q_{j,0}^{\smiley}(x_{j,0}, a_{j,0}) = R^{(j)} < \infty$, then from Lemma 5,

$$\left| R^{(j)} - Q_0^*(x_{j,0}, a_{j,0}) \right| = \left| Q_{j,0}^{\smiley}(x_{j,0}, a_{j,0}) - Q_0^*(x_{j,0}, a_{j,0}) \right| \leq 2\rho H.$$

Thus, $R^{(j)} \geq Q_0^*(x_{j,0}, a_{j,0}) - 2\rho H$. Furthermore, from Eqn(11), $Q_0^*(x_{j,0}, a_{j,0}) \geq V_0^*(x_{j,0}) - 4\rho H$. Thus we have $R^{(j)} \geq V_0^*(x_{j,0}) - 6\rho H$, and hence

$$V_0^*(x_{j,0}) - R^{(j)} \leq 6\rho H = O(\rho H).$$

**q.e.d.**

Thus, Proposition 1 directly follows from Lemma 6. Before proving Theorem 4, we first define some useful notations. Specifically, for any $j = 0, 1, \cdots$, we define $t_j^*$ as the last period $t$ in episode $j$ s.t. $Q_{j,t}^{\smiley}(x_{j,t}, a_{j,t}) = \infty$. If $Q_{j,t}^{\smiley}(x_{j,t}, a_{j,t}) < \infty$ for all $t = 0, 1, \cdots, H-1$, we define $t_j^* = $ NULL. We then have the following lemma:

**Lemma 7** $\sum_{j=0}^{\infty} \mathbf{1}[t_j^* \neq \text{NULL}] \leq K$, where $K$ is the number of partitions.

**Proof:**

$\forall j = 0, 1, \cdots$, if $t_j^* \neq \text{NULL}$, then by definition of $t_j^*$, $Q_{j,t_j^*}^{\odot}(x_{j,t_j^*}, a_{j,t_j^*}) = \infty$. We now show that $Q_{j',t_j^*}^{\odot}(x_{j,t_j^*}, a_{j,t_j^*}) < \infty$ for all $j' > j$, and $Q_{j',t_j^*}^{\odot}(x_{j,t_j^*}, a_{j,t_j^*}) = \infty$ for all $j' \leq j$.

Assume that $(x_{j,t_j^*}, a_{j,t_j^*}, t_j^*)$ belongs to partition $\mathcal{Z}_{t_j^*,k}$, thus $Q_{j',t_j^*}^{\odot}(x_{j,t_j^*}, a_{j,t_j^*}) = \overline{\theta}_{t_j^*,k}^{(j')}$, $\forall j' = 0, 1, \cdots$. Based on our discussion at the beginning of this section, $\overline{\theta}_{t_j^*,k}^{(j')}$ is monotonically non-increasing in $j'$. Thus, $Q_{j',t_j^*}^{\odot}(x_{j,t_j^*}, a_{j,t_j^*})$ is monotonically non-increasing in $j'$, and hence for any $j' \leq j$, we have $Q_{j',t_j^*}^{\odot}(x_{j,t_j^*}, a_{j,t_j^*}) = \infty$. Furthermore, to prove that $Q_{j',t_j^*}^{\odot}(x_{j,t_j^*}, a_{j,t_j^*}) < \infty$ for all $j' > j$, it is sufficient to prove that

$$Q_{j+1,t_j^*}^{\odot}(x_{j,t_j^*}, a_{j,t_j^*}) < \infty.$$

From OCP, the algorithm will add a new constraint $L_{j,t_j^*} \leq Q_{t_j^*}(x_{j,t_j^*}, a_{j,t_j^*}) \leq U_{j,t_j^*}$. We first prove that $U_{j,t_j^*} < \infty$. To see it, notice that if $t_j^* = H - 1$, then $U_{j,t_j^*} = U_{j,H-1} = R_{H-1}(x_{j,H-1}, a_{j,H-1}) < \infty$. On the other hand, if $t_j^* < H - 1$, then by definition

$$U_{j,t_j^*} = R_{t_j^*}(x_{j,t_j^*}, a_{j,t_j^*}) + \sup_{a \in \mathcal{A}} Q_{j,t_j^*+1}^{\odot}(x_{j,t_j^*+1}, a) = R_{t_j^*}(x_{j,t_j^*}, a_{j,t_j^*}) + Q_{j,t_j^*+1}^{\odot}(x_{j,t_j^*+1}, a_{j,t_j^*+1}).$$

From the definition of $t_j^*$, $Q_{j,t_j^*+1}^{\odot}(x_{j,t_j^*+1}, a_{j,t_j^*+1}) < \infty$, thus $U_{j,t_j^*} < \infty$.

Based on our discussion at the beginning of this section, due to the constraint selection algorithm, for episode $j + 1$, we have

$$Q_{j+1,t_j^*}^{\odot}(x_{j,t_j^*}, a_{j,t_j^*}) = \overline{\theta}_{t_j^*,k}^{(j+1)} = \min\{\overline{\theta}_{t_j^*,k}^{(j)}, U_{j,t_j^*}\} \leq U_{j,t_j^*} < \infty.$$

Thus, $Q_{j+1,t_j^*}^{\odot}(x_{j,t_j^*}, a_{j,t_j^*}) < \infty$ and hence $Q_{j',t_j^*}^{\odot}(x_{j,t_j^*}, a_{j,t_j^*}) < \infty$ for all $j' > j$.

Thus, if we consider $Q_{j',t_j^*}^{\odot}(x_{j,t_j^*}, a_{j,t_j^*}) = \overline{\theta}_{t_j^*,k}^{(j')}$ as a function of $j'$, then this function transits from infinity to finite values in episode $j$. In summary, $t_j^* \neq \text{NULL}$ implies that $\overline{\theta}_{t_j^*,k}^{(j')}$ transits from infinity to finite values in episode $j$. Since other $\overline{\theta}_{t,k}^{(j')}$'s might also transit from $\infty$ to finite values in episode $j$, we have

$$\mathbf{1}[t_j^* \neq \text{NULL}] \leq \ \# \text{ of } \overline{\theta}_{t,k}^{(j')}\text{'s transiting from } \infty \text{ to finite values in episode } j, \quad \forall j = 0, 1, \cdots.$$

Note that from the monotonicity of $\overline{\theta}_{t,k}^{(j')}$, for each partition, this transition can occur at most once, and there are $K$ partitions in total. Hence we have

$$\sum_{j=0}^{\infty} \mathbf{1}[t_j^* \neq \text{NULL}] \leq K.$$

**q.e.d.**

Finally, we prove Theorem 4.

**Proof for Theorem 4:**

First, notice that $\forall j = 0, 1, \cdots$, if $t_j^* = \text{NULL}$, then by definition of $t_j^*$, $\forall t = 0, 1, \cdots, H - 1$, $Q_{j,t}^{\odot}(x_{j,t}, a_{j,t}) < \infty$. Then from Lemma 6, we have

$$V_0^*(x_{j,0}) - R^{(j)} \leq 2\rho H(H + 1).$$

Thus, $V_0^*(x_{j,0}) - R^{(j)} > 2\rho H(H+1)$ implies that $t_j^* \neq \text{NULL}$, which is equivalent to

$$\mathbf{1}\left[V_0^*(x_{j,0}) - R^{(j)} > 2\rho H(H+1)\right] \leq \mathbf{1}\left[t_j^* \neq \text{NULL}\right], \quad \forall j = 0, 1, \cdots.$$

Thus, we have

$$\sum_{j=0}^{\infty} \mathbf{1}\left[V_0^*(x_{j,0}) - R^{(j)} > 2\rho H(H+1)\right] \leq \sum_{j=0}^{\infty} \mathbf{1}\left[t_j^* \neq \text{NULL}\right] \leq K,$$

where the last inequality follows from Lemma 7. Thus,

$$|\{j : R^{(j)} < V_0^*(x_{j,0}) - 2\rho H(H+1)\}| \leq K.$$

**q.e.d.**

Finally, we prove Corollary 2:

**Proof for Corollary 2:**
Notice that by definition, we have

$$\text{Regret}(T) = \sum_{j=0}^{\lfloor T/H \rfloor - 1}\left[V_0^*(x_{j,0}) - R^{(j)}\right].$$

Thus we have

$$
\begin{aligned}
\text{Regret}(T) &= \sum_{j=0}^{\lfloor T/H \rfloor - 1}\left[V_0^*(x_{j,0}) - R^{(j)}\right]\mathbf{1}\left[V_0^*(x_{j,0}) - R^{(j)} > 2\rho H(H+1)\right] \\
&+ \sum_{j=0}^{\lfloor T/H \rfloor - 1}\left[V_0^*(x_{j,0}) - R^{(j)}\right]\mathbf{1}\left[V_0^*(x_{j,0}) - R^{(j)} \leq 2\rho H(H+1)\right] \\
&\leq 2\overline{R}H\sum_{j=0}^{\lfloor T/H \rfloor - 1}\mathbf{1}\left[V_0^*(x_{j,0}) - R^{(j)} > 2\rho H(H+1)\right] \\
&+ 2\rho H(H+1)\sum_{j=0}^{\lfloor T/H \rfloor - 1}\left[V_0^*(x_{j,0}) - R^{(j)}\right] \\
&\leq 2\overline{R}HK + 2\rho H(H+1)\lfloor T/H \rfloor \leq 2\overline{R}KH + 2\rho(H+1)T.
\end{aligned}
$$

**q.e.d.**

# Appendix D   Proof for Proposition 2

We now prove Proposition 2.

**Proof for Proposition 2:**
If $\mathcal{Q}$ is a linear subspace/polytope with dimension $d$, then in one period, OCP needs to perform the following computation:

1. Construct $\mathcal{Q}_{\mathcal{C}}$ by constraint selection algorithm. This requires sorting $|\mathcal{C}|$ constraints by comparing their upper bounds and positions in the sequence (with $O\left(|\mathcal{C}|\log|\mathcal{C}|\right)$ operations), and checking whether $\mathcal{Q}_{\mathcal{C}} \cap \mathcal{C}_{\tau} \neq \varnothing$ for $|\mathcal{C}|$ times. Note that checking whether $\mathcal{Q}_{\mathcal{C}} \cap \mathcal{C}_{\tau} \neq \varnothing$ requires solving an LP feasibility problem with $d$ variables and $O\left(|\mathcal{C}|\right)$ constraints.

2. Choose action $a_{j,t}$. Note that $\sup_{Q \in \mathcal{Q}_{\mathcal{C}}} Q_t(x_{j,t}, a)$ can be computed by solving an LP with $d$ variables and $O\left(|\mathcal{C}|\right)$ constraints, thus $a_{j,t}$ can be derived by solving $|\mathcal{A}|$ such LPs.

3. Compute the new constraint $L_{j,t} \leq Q_t(x_{j,t}, a_{j,t}) \leq U_{j,t}$. Note $U_{j,t}$ can be computed by solving $|\mathcal{A}|$ LPs with $d$ variables and $O\left(|\mathcal{C}|\right)$ constraints, and $L_{j,t}$ can be computed by solving one LP with $d$ variables and $O\left(|\mathcal{C}| + |\mathcal{A}|\right)$ constraints.

If we assume that all the encountered numerical values can be represented with $B$ bits, and use Karmarkar's algorithm to solve LPs, then for an LP with $d$ variable and $m$ constraints, the number of bits input to Karmarkar's algorithm is $O\left(mdB\right)$, and hence it requires $O\left(mBd^{4.5}\right)$ operations to solve the LP. Thus, the computational complexities for the first, second, third steps are $O\left(|\mathcal{C}|^2 d^{4.5} B\right)$, $O\left(|\mathcal{A}||\mathcal{C}|d^{4.5}B\right)$ and $O\left(|\mathcal{A}||\mathcal{C}|d^{4.5}B\right)$, respectively. Hence, the computational complexity of OCP is $O\left(\left[|\mathcal{A}| + |\mathcal{C}|\right]|\mathcal{C}|d^{4.5}B\right)$ operations per period. **q.e.d.**

## Footnotes

[1]In general, the RL algorithm can choose actions randomly. If so, all the results in this section hold on the realized sample path.

[2]More precisely, in this constructive proof, the adversary does not need to adaptively choose the system function $F$. He can choose $F$ beforehand.