[Reviews · NeurIPS 2013]

Submitted by Assigned_Reviewer_2

The paper introduces a new reinforcement learning method for learning the optimal policy in deterministic undiscounted MDPs. The method makes use of the celebrated "optimism in the face of uncertainty principle" to strike a balance between exploration and exploitation. The result shows that in the case of deterministic episodic MDPs it is possible to achieve a bounded regret which only depends on some measure of the complexity of the function class used for estimating the value function. The main idea is to maintain upper and lower bounds on the value function of the states encountered at each episode and then use these bounds as the constraints for the value function in the next episodes.


The paper is very well written and easy to follow. Also I found the idea of constraint propagation very interesting and intuitive. Though I think the authors can do a better job in motivating the high level ideas behind this approach (probably in a longer version). Also the hardness result of Theorem 2 which shows that for a class of problems the upper bound is sharp adds to the value of this result.

-My only major concern (if any) is that this method relies on the assumption that the optimal value function exists in the hypothesis class which is a kind of restrictive assumption and usually is not the case in the real-world problems. The paper has proposed a solution for the "agnostic case" which does not rely on this assumption. However this result only applies to the state-aggregation kind of function approximators. Also the regret in this case grows exponentially w.r.t. the state dimension which makes this algorithm intractable for high-dimensional problems (note that typically K=O(k^d) where k is the number partitions for each dimension).

- The paper fails to recognize some of the recent related work:

1- The idea of minimizing the regret in continuous state-action problems in a more general stochastic setting has been addressed in the recent work by [Ortner and Ryabko 2012]. Similar to section 4.3 of this paper [Ortner and Ryabko 2012] rely on the state-aggregators to estimate the optimal value function.

2- There are some similarities between the work of this paper and the recent work of [Lattimore et. al. 2013], since both approaches rely on the idea of maintaining lower and upper bounds on the value functions and try to tighten this interval at each episode. Also They don't have the explicit dependency on the number of states and actions in their bounds, rather their results depend on the complexity of the hypothesis class


3- With regard to the policy-based algorithms the paper cites a relatively old reference that does not include some recent advancement in the policy search literature ( see e.g., [Scherrer and Geist 13] which proves that the policy search method can achieve the global maxima under certain assumption and [Azar et. al. 13] which proves regret bounds for a policy search algorithm with no dependency on the size of state-action space).

References:

[Ortner and Ryabko 12] R. Ortner, D. Ryabko, "Online Regret Bounds for Undiscounted Continuous Reinforcement Learning", NIPS (2012).

[Latimore et. al. 13] T. Lattimore, M. Hutter and P. Sunehag. "The Sample-Complexity of General Reinforcement Learning", ICML (2013).

[ Scherrer and Geist 13 ] B. Scherrer and M. Geist "Policy Search: Any Local Optimum Enjoys a Global Performance Guarantee." INRIA Tech report (2013).

[Azar et. al. 13] M. Azar, A. Lazaric and E. Brunskill " Regret bounds for reinforcement learning with policy advice." ArXiv Report, (2013).
Summary: The paper introduces a new RL algorithm for large-scale deterministic systems. Sharp performance guarantees in the form of upper and lower regret bounds have been provided.

Submitted by Assigned_Reviewer_4

*Summary*
This paper proposes an algorithm for episodic reinforcement learning (RL) in deterministic environments when the learner has a set of possible Q-functions at his disposal. Sample complexity and regret bounds are given that depend on the so-called "eluder dimension" of the function set and are independent of other parameters. The agnostic case is considered as well.

*Evaluation*
There is a lot to like about this paper. It is original, well-written, and has some nice-looking results. On the other hand, the setting the paper considers is very simple (episodic learning, deterministic environment), and it is questionable whether the results can be generalized to more interesting settings. Further, there are also a few things that I found improvable:

- With the exception of the mentioned linear systems with quadratic cost, I found it not very natural to have a set of possible Q-functions at the disposal of the learner. Thus, the results for the "tabula rasa" case and agnostic learning seem to be the more important ones. However, for the tabula rasa case, it seems one does not get anything better than when doing pure exploration until all states have been visited. (I think this deserves to be mentioned.) Concerning agnostic learning, in Theorem 4 there is an unresolved dependence between K and rho, so that it is not clear what can be really accomplished in this case.

- It would be good to include at least some proof sketch for the main theorem to give a more detailed idea why the algorithm works. There is enough space for this, even more when the special hypothesis classes are dealt with more concisely. In particular, the lines 282-295 are simply redundant.

- Finally, some related work should be mentioned w.r.t.
* RL with a set of possible models: See recent work of Marcus Hutter and co-authors, and the paper of Maillard et al at NIPS 2011 and two subsequent papers by the same author and various co-authors at ICML and AISTATS this year.
* Deterministic RL: Bernstein & Shimkin, COLT 2008.


*Minor comments*
- In l.47, I found it not clear what is precisely meant by "model-based algorithms".
- In l.82, according to the results near-optimality is not so clear, cf. the remark on the dependence of K and rho in Theorem 4 above. The sentence as stated gives the wrong impression that one gets better results with fewer basis functions.
- In l.124, I did not see why R^(j) is strictly smaller than V*_0(x_j,0). It seems, it should be \leq instead.
- In lines 125 and 139 L is used with different meaning. It would be better to use two different letters instead.
- In l.144 "more recent experience" assumes that there is some temporal order in the constraints. This should be made clear before.
- In l.168 in Algorithm 2, it should be made clear that the set C comes from Algorithm 1.
- In l.177, \cap should be used for set intersection.
- In l.209, "transition" is used as a verb, which I think is not correct.
- In l.229, "Q to denote" should be "Q denote".
- In l.339, it was not clear to me why the condition in Proposition 1 means that the constraints are "redundant" (and not just consistent).

---

Remarks to author's response:

- Pure exploration: What I meant is that since the system is deterministic, one could first explore till the whole system is known and then exploit.

- Payoff between K and rho: If one chooses a finer discretization (higher K) then the error rho will usually be smaller. Hence there seems to be a payoff between K and rho in Theorem 4, and it is not clear what one can expect for an optimal discretization.
Summary: An original, well-written paper with some nice results, which however are probably difficult to apply or generalize.

Submitted by Assigned_Reviewer_5

This paper presents an approach to the episodic and deterministic RL problem under function approximation, providing both computational and sample complexity guarantees under certain assumptions.

While the paper is well-written and has theoretical interest, I think the authors miss a significant part of the literature in this area, which should be incorporated in order to make the paper publishable:
- in line 31 they should cite the fairly extensive KWIK literature starting in 2008.
- in line 48 they mention model-based algorithms with provable sample complexity (they correctly mention the intractable planning problem), and again they should cite some relevant work in the KWIK framework (Walsh, Szita, Diuk, Littman 2009; Diuk, Li, Leffler 2009; Brunskill, Leffler, Li, Littman and Roy 2009; among others).
- in lines 63-64 they talk about a lack of bounds in the context of function approximation, and I think Li and Littman 2008 and 2010 should be cited.
- in lines 67-68, the authors should refer to an important literature that provides theoretical bounds for planning. There is a lot of work coming out of Remi Munos' group on this, as well as from the planning community. These two papers from the Munos group come to mind: http://uai.sis.pitt.edu/papers/07/p67-coquelin.pdf and http://colt2010.haifa.il.ibm.com/papers/14Bubeck.pdf. Also, I think Walsh, Goschin and Littman 2010 should be cited as a reference that combines efficient learning with sample-based planning.
- in lines 90-91, add to the literature on agnostic RL a reference to Szita and Csepesvapari 2011 (from COLT), which is also relevant here.

The authors correctly claim that existing efficient learning algorithms that assume an oracle planner are limited in their practical interest (although many approximate planning methods can be used and effectively integrated, as in Walsh, Goschin and Littman 2010). If OCP overcomes this issue and becomes interesting in practice and not only theoretically, the authors should demonstrate it through some experiments in interesting domains.

As for the theoretical appeal of OCP, once again in section 4.1, when analyzing the complexity under different classes of systems, the paper would be enriched by comparing it against theoretical guarantees in the existing literature.

In summary, I think this paper could be dramatically improved by covering the existing literature more thoroughly, as well as by demonstrating the efficacy of the method through some examples.

Minor comments:
- line 39: import -> importance
- line 101: in definition of MDP, why use F and not the more standard T?
- Section 3: it is not clear how Algorithms 1 and 2 get integrated. As I understand, whenever Alg 2 refers to Q_C, this gets computed by Alg 1, is that right?
Summary: The paper is well-written and has theoretical interest, but it fails to cite important literature in this area. It could also be improved by including experiments.
Author Feedback

Author rebuttal: We thank all reviewers for their most insightful comments. We very much appreciate that Reviewer 2 and 4 found the paper worthy of an accept recommendation, and all the reviewers thought the paper is well written. We also would like to thank all the reviewers for pointing out some relevant literature and typos in the paper. We will fix the typos and cite the related literature.

To Reviewer 2:

We agree that adding a paragraph on the high level ideas behind the OCP algorithm will improve the paper and plan to do so.

As to your major concern, it is worth pointing out that the OCP algorithm can be applied as a general agnostic learning method, and we believe the algorithm is effective in such contexts. However, efficiency analysis in this broader context remains an open issue. Our current paper presents efficiency results limited to the “coherent hypothesis class” and “state aggregation” contexts. It is worth noting that, though they are restrictive, these contexts capture tabula rasa RL, LQ control, and general state aggregation schemes (where the number of partitions need not grow exponentially in the state space dimension).

To Reviewer 4:

We will include proof sketches for the main theorems of the paper to provide more explanation on why OCP works.

As to the sample complexity in the tabula rasa case, we are not sure what you mean by “pure exploration.” However, it is worth pointing out that even for the tabula rasa case, some classical exploration schemes (e.g. epsilon-greedy exploration, Boltzmann exploration) can give rise to sample complexity that grows exponentially in |S|, the cardinality of the state space. On the other hand, the sample complexity of OCP in the tabula rasa case is linear in |S|.

Regarding agnostic learning, note that for a given environment, once the state aggregation scheme is chosen, both K and rho are uniquely determined. Thus, we are not quite sure what you mean by “unsolved dependence between K and rho.”

To Reviewer 5:

Thanks a lot for pointing us to several papers. After carefully going through the recommended literature, we believe that Li and Littman 2008 and 2010 are directly related to our paper, and some of the other papers are also relevant though less directly related. We will revise the paper to discuss representative literature on model-based algorithms in the KWIK framework, Li and Littman 2008 and 2010, and Szita and Csepesvari 2011. We do not think the literature on planning is as relevant.

It is worth pointing out that the contributions of our paper are novel and significant relative to the additional citations. Specifically, all the papers except Li and Littman 2008 and 2010 are concerned with different RL settings (most of them focus on provably efficient model-based RL algorithms, while others are concerned with planning). Model-based RL algorithms require planning (i.e., dynamic programming) to choose actions, and this makes such algorithms computationally intractable in the face of large-scale models. Li and Littman 2008 and 2010 consider a similar setting as our paper (model-free RL with compact value function approximation (VFA)), in the more general stochastic transition setting. They propose a RL algorithm (REKWIRE) in the KWIK framework, and show that if the KWIK online regression problem can be solved efficiently, then the sample complexity of REKWIRE is polynomial. However, they do not show how to solve general KWIK online regression problem efficiently (i.e. with a polynomial sample complexity), and it is also not clear if it is possible. Thus, Li and Littman 2008 and 2010 do not provide a complete provably sample efficient RL algorithm for the VFA setting, even for the simplest “coherent hypothesis class” case. The authors correctly describe their contributions as “reducing RL to KWIK online regression”.

Though our paper only focuses on RL in deterministic systems, we propose a complete RL algorithm based on VFA (OCP), which is provably sample efficient in the “coherent hypothesis class” case and “state aggregation” case. To the best of our knowledge, it is the first complete provably sample efficient RL algorithms in these two cases. Further, for the “coherent hypothesis class” case, we also provide lower bounds on the sample complexity and regret, and show the sample complexity and regret of OCP match these lower bounds.

We agree that OCP should be tested through experiments in interesting domains. We plan to report experimental results in future work. We will also compare the efficiency of OCP and other algorithms for different classes of systems (e.g. LQ).

As to your minor comments:
(1) F is a standard notation for deterministic transition functions;
(2) Your understanding of how Algorithms 1 and 2 are integrated is correct.